# Synthesizing world models for bilevel planning

**Zergham Ahmed**  *zerghamahmed@g.harvard.edu*
*Department of Computer Science*
*Harvard University*

**Joshua B. Tenenbaum**  *jbt@mit.edu*
*Department of Brain and Cognitive Sciences*
*MIT*

**Christopher J. Bates**[†]  *cbates@ihmc.org*
*Institute for Human and Machine Cognition, Harvard University*

**Samuel J. Gershman**[†]  *gershman@fas.harvard.edu*
*Department of Psychology and Center for Brain Science*
*Harvard University*

[†]*Equal advising*

**Reviewed on OpenReview:** *https://openreview.net/forum?id=m9V4JHLJrD*

## Abstract

Modern reinforcement learning (RL) systems have demonstrated remarkable capabilities in complex environments, such as video games. However, they still fall short of achieving human-like sample efficiency and adaptability when learning new domains. Theory-based reinforcement learning (TBRL) is an algorithmic framework specifically designed to address this gap. Modeled on cognitive theories, TBRL leverages structured, causal world models—"theories"—as forward simulators for use in planning, generalization and exploration. Although current TBRL systems provide compelling explanations of how humans learn to play video games, they face several technical limitations: their theory languages are restrictive, and their planning algorithms are not scalable. To address these challenges, we introduce TheoryCoder, an instantiation of TBRL that exploits hierarchical representations of theories and efficient program synthesis methods for more powerful learning and planning. TheoryCoder equips agents with general-purpose abstractions (e.g., "move to"), which are then grounded in a particular environment by learning a low-level transition model (a Python program synthesized from observations by a large language model). A bilevel planning algorithm can exploit this hierarchical structure to solve large domains. We demonstrate that this approach can be successfully applied to diverse and challenging grid-world games, where approaches based on directly synthesizing a policy perform poorly. Ablation studies demonstrate the benefits of using hierarchical abstractions.

## 1 Introduction

Video games are a useful testbed for reverse engineering human intelligence. They occupy a middle ground between overly simple and complex environments, allowing systematic study of key engineering principles. Reinforcement learning (RL) has demonstrated impressive capabilities in games. Model-free RL, which has its roots in basic animal learning processes (Sutton & Barto, 1981; Schultz et al., 1997) has been successful at achieving superhuman performance on Atari games (Mnih et al., 2015). However, research comparing human and machine learning in Atari games has shown that, even when matched for performance level, humans learn orders of magnitude faster than these state-of-the-art deep RL algorithms (Tsividis et al.,

2017). Furthermore, these RL algorithms fail to generalize, even when faced with small variations of the game (Kansky et al., 2017).

More recently, AI researchers have explored how "foundation" models, such as large language models (LLMs), can be used to build game-playing agents. Foundation models can act as powerful, general purpose reasoners in many domains, since they contain vast amounts of general-purpose and even game-specific world knowledge. However, they fail catastrophically in situations where examples cannot be given or the domain is too far from their training set. In addition, current transformer-based models struggle with many kinds of planning that are essential to performing well in games (Ruoss et al., 2025; Paglieri et al., 2024; Waytowich et al., 2024; Kambhampati et al., 2024).

Humans seem to learn and plan in fundamentally different ways than both foundation models and end-to-end RL models (Tsividis et al., 2017; Pouncy et al., 2021; Pouncy & Gershman, 2022). Building on Tsividis et al. (2021), the goal of our work is to develop algorithms inspired by how humans leverage their prior knowledge to learn new games with high sample efficiency. In particular, Tsividis et al. (2021) argued that humans learn structured causal models ("theories") of complex domains like video games, and then use these theories for planning and exploration. This form of *theory-based reinforcement learning* (TBRL) is inspired by evidence that humans acquire and use intuitive theories from a young age (Gopnik & Meltzoff, 1997; Gerstenberg & Tenenbaum, 2017), revising their theories through exploration and manipulation of the environment. TBRL contrasts with other model-based approaches in that agents explicitly represent the kinds of rich causal and relational structures that humans appear to reason over (for example, sprites in a game and their collision rules). Existing model-based RL systems learn effectively in tasks that emphasize motor skills and precise timing, but struggle to learn conceptually challenging domains requiring extensive exploration (Hafner et al., 2023). Work on TBRL suggests that this gap can be closed by learning world models which more closely resemble human intuitive theories.

Tsividis et al. (2021) established a proof of concept for TBRL with their Exploring, Modeling, and Planning Agent (EMPA), which they applied to a suite of video games. EMPA learns a theory of a game's rules expressed in a domain-specific language called the Video Game Description Language (VGDL), which is capable of representing a wide variety of Atari-style video games (Schaul, 2013). EMPA's use of VGDL was argued to roughly capture the kinds of causal, object-oriented representations that humans reason over intuitively. Learned theories are then used as forward simulators, allowing a search algorithm to plan and explore more effectively. EMPA learned at approximately the same rate as humans in many games, in stark contrast to state of the art deep RL methods.

Scaling TBRL remains challenging, however. Current implementations of TBRL systems face two major limitations. First, their theory languages are restricted to domains that can be represented in VGDL. Humans, by contrast, can evidently build not only VGDL-style theories but many other kinds of theories as well. Second, the planning algorithms used in previous TBRL agents are not scalable to large state spaces. This is because they rely on value-based iteration, which suffers from the curse of dimensionality when computing the value for large state-action pairs (Sutton & Barto, 1998).

Our work extends TBRL with new algorithmic techniques that allow agents to be more skilled and flexible learners. Our agents make learning both more flexible and more tractable by reusing previously learned, high-level symbolic abstract concepts, adapting them to the particulars of a new domain via a fast-learning mechanism based on program synthesis. These high-level abstractions include planning operators that can dramatically speed planning and support efficient exploratory actions. If an agent can fill in these low-level details quickly in new contexts, then it can recycle useful concepts without having to relearn them from scratch every time. Second, our approach can acquire a much richer set of world models beyond VGDL by writing code in a highly expressive language (specifically, Python). This is made possible by leveraging large language models (LLM) for fast and general program synthesis.

We introduce TheoryCoder[1], an agent that generalizes quickly to new problem domains by leveraging prior knowledge in the form of general-purpose, high-level abstractions (expressed in the Planning Domain Definition Language, or PDDL; Ghallab et al., 1998; McDermott, 2000). TheoryCoder grounds these abstractions

---

[1]Code is available at https://github.com/ZerghamAhmed/TheoryCoder.

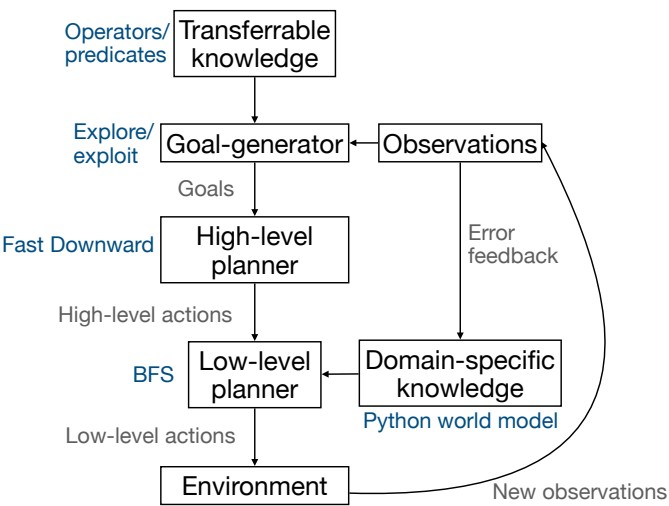

Figure 1: **TheoryCoder overview**. The agent uses high-level abstractions (assumed to be previously learned), in the form of PDDL operators and predicates, as its transferable knowledge to do high-level planning and sample exploratory goals. TheoryCoder learns a new domain by inferring how high-level concepts ground out in low-level actions (a world model specified in Python code, synthesized by prompting a large language model). Using its learned world model, it finds a sequence of low-level actions that it predicts will satisfy the high-level plan. The bilevel planner integrates a high-level component (Fast Downward) which uses the PDDL abstractions, and a low-level component (breadth-first search, BFS) which uses the learned world model. If there are errors in its predictions, TheoryCoder refines its world model.

in particular environments by synthesizing a low-level transition model using LLM queries. The transition model is highly flexible, as it can represent any dynamics expressible in Python, yet learning is made tractable in part because it is anchored on high-level concepts, which constrain decisions about how to organize the code. TheoryCoder then uses the learned model and high-level abstractions in a fast bilevel planning algorithm to solve challenging tasks that require various forms of reasoning. The bilevel planner harnesses the abstract PDDL operators to constrain search in the low-level state space, similar to the kinds of hierarchical planning strategies used by humans (Tomov et al., 2020; Correa et al., 2025). We show that this approach can be successfully applied to complex domains, such as the video game Baba is You, substantially outperforming baselines that use LLMs directly for planning. In particular, TheoryCoder achieves higher success rates, higher sample efficiency, and better generalization.

## 2 Related Work

### 2.1 Theory-based RL

#### 2.1.1 Human Learning

Humans are able to leverage their prior knowledge to make inferences that go far beyond the available data, allowing them to generalize to novel domains with only sparse observations. It has been proposed that humans are able to do this via strong inductive biases which constrain the vast hypothesis space (Tenenbaum et al., 2011; Gershman, 2021). In particular, human inductive biases often take the form of structured "intuitive theories" which share some characteristics with scientific theories: they are causal, abstract, domain-specific, and explanatory (Gopnik & Meltzoff, 1997; Gerstenberg & Tenenbaum, 2017). Intuitive theories explain observational data by reference to underlying causal laws that generalize across many superficially different events. The causal laws are different for physics, biology, psychology, and other domains. For example, intuitive theories of physics explain the dynamics of objects in terms of mechanics, whereas intuitive theories of psychology explain the dynamics of agents in terms of beliefs, utilities, and plans.

The mature network of intuitive theories found in adults places a strong constraint on what kinds of tasks can be learned with more or less difficulty. The key point for our purposes is that human learning of complex tasks like video games can be understood as a form of domain-specific theory learning, subject to abstract structural constraints and prior knowledge. A number of empirical studies have supported this claim, finding that human learning is best described as a process of theory induction (Pouncy et al., 2021; Pouncy & Gershman, 2022), accompanied by neural representations of theories in prefrontal cortex (Tomov et al., 2023).

#### 2.1.2 Human learning principles missing in Deep RL

These principles are not currently incorporated into popular RL systems. For example, model-free deep RL methods like the Deep Q-learning Network (Mnih et al., 2015) and its descendants have relatively weak inductive biases. They can, with extensive training, learn to perform at or above human performance on many challenging tasks (e.g., Atari games), but often exhibit brittle generalization and poor sample efficiency on new tasks (Tsividis et al., 2017; 2021).

Classical model-based RL systems learn models of an environment, represented as state-action transition tables (Sutton & Barto, 1998), which do not scale to large state spaces. Other methods use deep neural networks to approximate the dynamics (Campbell et al., 2019; Zhang et al., 2019; Schrittwieser et al., 2020). MuZero (Schrittwieser et al., 2020) is a notable example of a model-based RL agent, which outperformed model-free competitors in Atari games and other domains. Its advantage may have been related to its ability to learn a compressed, abstract model of the state transition function (through end-to-end training). However, MuZero and other neural model-based approaches do not explicitly represent causal laws governing interactions between objects, which limits their ability to generalize effectively. TBRL is committed to a stronger set of inductive biases for model learning: dynamics are modeled in terms of object-oriented causal laws.

### 2.1.3 Limitations of current Theory-Based RL systems

Despite the promise of TBRL, it faces several challenges related to learning with structured representations. First, in practice researchers end up hand-designing the agent's hypothesis space separately for each evaluation domain, since it is challenging to specify a single hypothesis space for all potential domains of interest. Our work here does not remove this problem, but it makes progress in addressing it. Specifically, the most notable previous instantiation of TBRL (EMPA), could only learn theories sampled from the VGDL grammar, which is restricted to simple collision-based interactions. Here we express our theories in part using Python, which is essentially unconstrained. Second, prior work learned theories using approximate Bayesian inference, which can be slow or intractable for large hypothesis spaces. Here, we take a neurosymbolic approach and leverage LLMs to do approximate inference.

Even once a theory has been learned, efficient planning is non-trivial. The run-time speed of EMPA, for example, is prohibitively slow for complex domains due to the cost of querying the theory-based simulator. We accelerate run-time computation by using bilevel planning with PDDL operators, as explained below.

## 2.2 Model-based reasoning

### 2.2.1 Neural Models

Model-based reasoning can be a powerful tool in many settings. For example, an agent who has an accurate model of their environment can use the same model to plan actions to reach a goal and to infer the most likely series of actions that produced an observation. Model-based reasoning has been studied in various contexts, including RL and robotics, and work in these areas have explored a number of techniques for building and representing models of an agent's environment. For instance, some studies have focused on neural world models that compress high dimensional state spaces (typically pixels) into a compact latent vector space(Hafner et al., 2023; Sehgal et al., 2024; Goyal et al., 2021; Ha & Schmidhuber, 2018). Representations in the latent space can then be used for simulation and prediction. For example, the RL agent can learn using fewer interactions with the environment by offline training using simulated rollouts from the current world model (Wu et al., 2023).

### 2.2.2 Symbolic Models

Our work is most closely aligned with a different family of approaches to model-learning, namely structured, symbolic world models. Most relevant within this family are program synthesis approaches, which seek to find programs that faithfully model key aspects of an environment's dynamics. Here, a learner's goal is to infer from a set of observations the program that generated it. A major advantage of using programs as world models is that they are structured and compositional. This property allows learners to more readily modify specific parts or aspects of a model in response to just one or a handful of new observations. By contrast, neural representations cannot easily be modified in this way.

A notable example of the program synthesis approach is AutumnSynth (Das et al., 2023), which learns dynamic, Atari-style games using a novel DSL (domain-specific langauge) called Autumn. Similarly, Evans et al. (2021) develop an approach that learns Datalog programs of domain dynamics from raw pixel input. Guzdial et al. (2017) is another work that learns symbolic rules for a game directly from raw pixel input. In all these types of works (including also EMPA), learners are able to construct precise world models that explain the observations well and support powerful reasoning. However, they face a fundamental challenge shared by all program synthesis techniques, which is the cost of search. Specifically, if the goal is to infer the program that produced a set of observations, it could be prohibitively expensive to enumerate all possible programs, and thus learners need ways to cut down search times. One popular solution is to design a DSL which has limited expressiveness and therefore only works for a narrow domain of interest. Such a DSL generates fewer possibilities to consider for a given program length, but the learner needs to possess many such DSLs in order to have broad, human-like competence across many domains.

### 2.2.3 Synthesizing models using LLMs

Recently, LLMs have provided a powerful tool for addressing this problem. LLMs do not synthesize code by explicitly enumerating over a space of programs. Rather, they can be seen as relying on a kind of amortized search—approximating the search *solution* without engaging in a search *process*. The approximation generally improves with the size of the training set; thus, LLMs excel at program synthesis due in part to their massive training sets. Leveraging this principle, Tang et al. (2024) demonstrated that LLMs can synthesize accurate world models in highly expressive languages (e.g., Python). Thus, LLM-based agents can learn to master a wider range of domains than in prior work.

Dainese et al. (2024) similarly demonstrated an ability to learn Python world models in a large variety of domains. However, they focus on an offline RL setting.

Building on these ideas, our approach also leverages LLMs for online program synthesis (specifically, synthesis of the low-level transition model). However, our work is unique in that our agents possess world models that live at two separate levels of abstraction, and this confers benefits to planning efficiency and learning convergence. The more abstract level of the world model (expressed in terms of PDDL operators) can be considered an abstract transition model, which guides and constrains learning of the low-level transition model and constrains search over plans, because it captures useful high-level structure in the domain that is not explicitly represented at the lower level.

### 2.3 Bilevel Planning

Long-horizon planning in tasks with large state spaces is difficult. Bilevel planning is one approach that has been used extensively for fast task-and-motion-planning in robotics (Garrett et al., 2021; Gravot et al., 2005). Bilevel planning uses abstract representations for states and actions. The abstract states are referred to as predicates, and the abstract actions are referred to as operators. Planning problems can be expressed in PDDL (Ghallab et al., 1998) using these operators and predicates. For a given planning problem, the domain and problem files represent the actions available in a state and the initial state of the task. A classical planner can be used to find a set of grounded operators that lead to the abstract goal state. Expressing a problem in PDDL enables the use of fast domain-independent classical planners (e.g., Helmert, 2006).

Abstractions are typically hand-designed instead of learned. Although there has been some work on learning abstractions (Silver et al., 2021; 2023; Kumar et al., 2023; Silver et al., 2022; Li & Silver, 2023), all of them have focused on relatively simple domains involving picking up and placing objects in long-horizon planning. Many of these methods rely on handcrafted components, typically the predicates. Works such as Wong et al. (2024) learn operators with few-shot demonstrations using an LLM to construct the predicates. Wong et al. (2024) use these operators in conjunction with a low-level controller that executes low-level actions in the domain. Our work differs in that we use the operators to guide the learning of a low-level transition model, which captures more granular domain mechanics, and use that low-level model to plan. Guan et al. (2023) leverage an LLM to learn the full PDDL domain, including predicates and operators, but use humans in the loop to refine them.

### 2.4 LLMs for planning and low-level policies

Past work has studied how an LLM can be used directly to plan in complex environments. The LLM is prompted to output actions by giving it a text-based state representation of the domain (Yao et al., 2022; Hao et al., 2023; Zhao et al., 2024; Liu et al., 2023) or by using VLLMs (Visual Large Language Models) (Waytowich et al., 2024; Paglieri et al., 2024; Ruoss et al., 2025; Cloos et al., 2024), which can be prompted with an image of the state. Various prompting strategies such as Chain-of-thought (CoT; Wei et al., 2022) are used to elicit better performance and more accurate plans. Ruoss et al. (2025) showed that even when given hundreds of expert multi-modal demonstrations in-context, these frontier models struggle at planning tasks in interactive environments. Many of these frontier models still suffer from spatial reasoning issues and hallucinations. Many of the mistakes they make are unlikely to be made by humans. Their performance is typically poor compared to humans, as they are only able to achieve 50% progress or less in most domains (Paglieri et al., 2024), even when using the most powerful closed-source models.

# 3 TheoryCoder

## 3.1 Overview

An iterative cycle of hierarchical planning, acting, and model updating is at the core of our TheoryCoder architecture (Fig. 1). It decomposes complex tasks into manageable high-level sub-tasks and continuously refines the world model based on feedback.

Fig. 2 illustrates how the PDDL planner generates high-level plans, which are then translated into executable actions by the low-level breadth-first search (BFS) planner. The BFS planner determines the appropriate action sequence by simulating transitions using the world model (Fig. 3). Once a viable sequence is identified, it is executed in the environment. If the high-level plan is not successfully achieved, the world model undergoes refinement, either by correcting explicit errors or identifying discrepancies through exploratory goals. If the simulation fails to terminate—indicating that no valid solution exists under the current world model—an exploratory goal is initiated to uncover missing dynamics in the model.[2]

## 3.2 Preliminaries

We adopt the definition of a classical planning problem, where transitions between states of the environment are specified by a transition function $T : \mathcal{S} \times \mathcal{A} \to \mathcal{S}$ with state space $\mathcal{S}$ and action space $\mathcal{A}$. The goal of the agent is to find a plan (sequence of actions) $\pi = <a_1, \ldots, a_N>$ that minimizes the sum cost $\sum_{n=1}^{N} c(s_n, a_n)$, where $c(s, a)$ is the cost of taking action $a$ in state $n$. Here we take $c(s, a) = K$ (a constant) for all non-goal states, and $c(s^*, a) = 0$ for the goal state $s^*$. An optimal plan is the shortest sequence of actions from $s_1$ to $s_N = s^*$ that reaches the goal.

We assume that the agent has access to an estimate $\hat{T} = f(D)$, where $D$ is its observation data and $f$ is an estimation algorithm (described further below).

## 3.3 Theory language

The transition function corresponds to the low-level (grounded) dynamics of the environment. The agent additionally represents abstract actions (operators) and states (predicates). An action abstraction $\omega(\pi)$ is a mapping from an action sequence to a target state; its inverse (from target state to action sequence, obtained by a planning algorithm) is denoted by $\phi^{-1}(s)$. A state abstraction $\phi(s)$ is a Boolean predicate that encodes features of the state that are important for planning. The combination of a low-level transition function and high-level abstractions together constitute the agent's domain theory.

Our system's theory language is composed of two programming languages: PDDL 1.2 (Ghallab et al., 1998) and Python, which together represent the domain hierarchically. PDDL represents the domain using high-level abstractions and handles the high-level planning for TheoryCoder. Python represents the domain's low-level state space and handles the low-level planning in the raw state space.

### 3.3.1 High-level planning and abstractions

TheoryCoder's abstract knowledge is encoded by PDDL domain and problem files, which describe different planning problems in a domain. The PDDL domain and problem files can be thought of as an abstract theory or a high-level transition function.

For a given planning problem, we first run Fast-Downward (Helmert, 2006), a PDDL planner, using the domain and problem file as input. Fast-Downward outputs a high-level plan in terms of the operators that were used in the PDDL domain file.

Once the system has the high-level plan, it converts this into a low-level action sequence that can be executed in the domain. Here, we utilize the low-level transition function written in Python for low-level planning. The low-level transition function (the granular theory of the domain) is the **only** component of our system that is learned; the problem file and domain files are hard-coded.

---

[2]We use "world model" and "transition model" interchangeably.

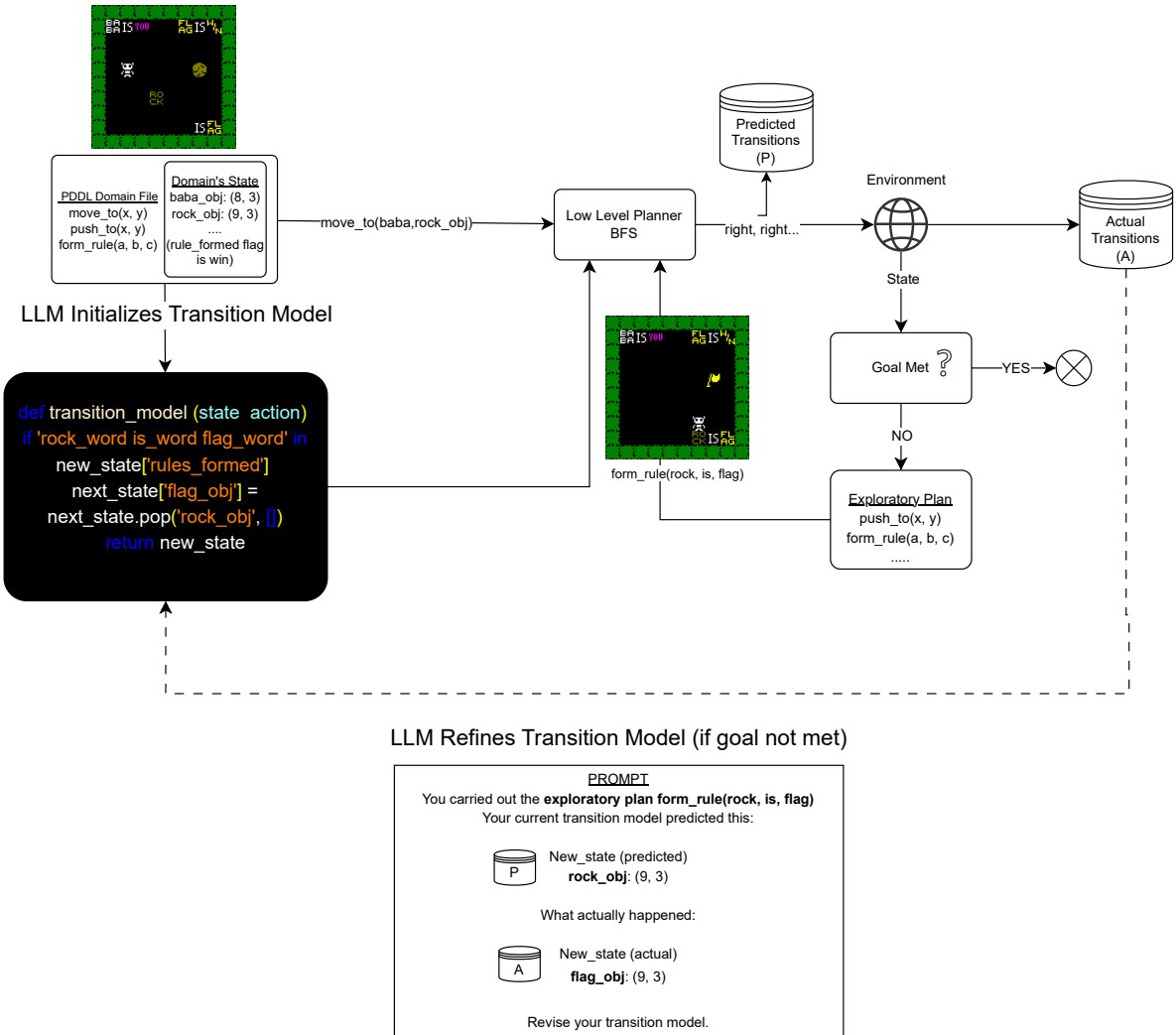

Figure 2: **TheoryCoder example**. A high level abstract plan is converted into a low-level action sequence. If the plan fails (implying an error in the transition model), an exploratory abstract subplan is carried out and the resulting replay buffer is used in a prompt to the LLM to refine the transition model. In this example, the replay buffer is labeled by the high-level abstraction "form_rule rock is flag". This gives the LLM a semantic clue in the form of natural language guidance about what part of the code to modify.

We contend that for the kinds of game domains we study in this paper, humans playing them for the first time already possess the kinds of abstract operators and predicates that we assume here.[3] The abstractions are analogous to the kinds of abstract causal laws that humans use to understand their environment: when we plan to go from one room to another, we conceptualize the plan in terms of abstract actions like "opening a door" rather than low-level actions like muscle commands. Likewise, we conceptualize the key events in this process through state abstractions like "door is locked" and "in the living room" rather than low-level states like detailed configuration of objects. The principal learning problem is grounding the abstractions in a particular domain. Once this grounding has been learned, the only remaining computational problem is planning. Future work will explore how an LLM can learn abstractions over longer time scales (e.g., multiple games, or multi-agent cultural transmission).

### 3.3.2 Low-level Python transition function and planning

We provide Python versions of the predicates which serve as a bridge between the high-level PDDL abstractions and low-level Python components of our system. Specifically, it is bridged by a function called "checker" that maps abstract states to low-level states. An abstract state can be described by a set of predicates. Recall that effects are a set of predicates that describe the new abstract state after a high-level subplan (a grounded operator) is executed. The checker function maps these effects to low-level states. In other words, the checker function determines whether a low-level state has all the effects satisfied for a given grounded operator. This state can be thought of as the goal or termination condition of the low-level BFS planner, which searches for the corresponding low-level actions for one grounded operator (one high-level subplan) at a time and then concatenates all the low-level actions at the end to return the overall low-level plan.

To illustrate, consider a simple 2D grid world where there is only an agent and a flag; the low-level action space consists of "up", "left", "right", and "down". The only operator for this domain is "move_to x y", which has the effect "overlapping x y". The high-level plan is "move_to agent flag". The low-level BFS planner finds the low-level actions corresponding to this plan. The only effect for "move_to agent flag" is the "overlapping agent flag". The low-level BFS planner will call the checker while searching each node (low-level state) to see whether that node satisfies the effect for "overlapping agent flag". After it finds a node where the effects are true, it will return the low-level action sequence. This example is illustrated in Fig. 3.

If the high-level plan has multiple subplans, then each subplan will be similarly handled: the BFS planner finds the low-level action sequences for each subplan, which are then concatenated to return the overall low-level action sequence.

### 3.4 Learning

Before the planning process is carried out, TheoryCoder takes (up to) 10 random actions to gather a replay buffer of experience. The LLM is then queried with a prompt that includes the initial state, the low-level executable action space, the replay buffer of random actions, and a description of the domain in natural language (see Appendix A for examples). The prompt asks the LLM to generate an initial Python low-level transition function.

During planning, TheoryCoder maintains its simulations in the replay buffer. Once planning is done, the low-level plan is executed in the environment and observed real transitions are added to the replay buffer. The simulated and actual plans in the replay buffer are used for theory refinement, as explained below.

If a successful low-level plan is not found, then there is likely an error in the low-level transition model (assuming the planner has been run long enough). In this case, TheoryCoder first analyzes the trajectory that it took to while following the high-level plan as shown in Fig. 4. The corresponding LLM response is shown in Fig. 10. It compares the predicted transitions and actual transitions to see if there are any mismatches. If there are, TheoryCoder refines the low-level transition function by sending the predicted and actual transitions to a prompt in the LLM, asking it to refine the transition model based on this mismatch.

---

[3]In other words, the abstractions are effectively "hard-coded" for humans at the time scale of gameplay.

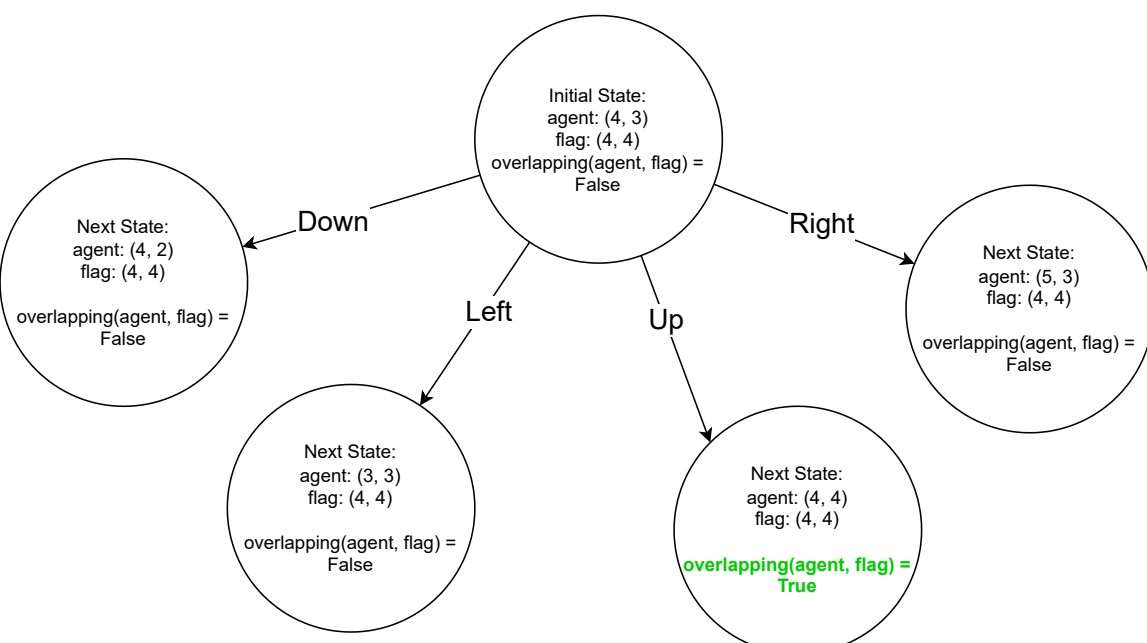

Figure 3: **PDDL High-level plan's connection to low-level planner**. For the high-level plan "move_to(agent, flag)" that has the effect "overlapping(agent, flag)", the low-level BFS planner searches for the low-level actions that bring the agent to a state where the effect is True. In this case, the "up" action leads to a low-level state where the effect is True, since the agent and flag's coordinates are the same. Note that during the BFS search, for each low-level action the low-level Python transition function is used to determine the next state.

If this is not sufficient to find the errors, then the error may be more nuanced. In this case, TheoryCoder enumerates some exploratory plans as shown in Fig. 5. The possible exploratory plans that are considered are based only on those high-level plans which have preconditions satisfied in the planning problem's initial abstract state. The complete process is summarized in Algorithm 1. The initialization and refinement prompt for the transition function is shown in Appendix A.

---

**Algorithm 1** TheoryCoder

---

**Input:** PDDL domain file $D$, PDDL problem file $\mathcal{P}$, LLM, initial state $s_0$, action space $\mathcal{A}$
**Output:** Low-level plan $\pi = \langle a_1, a_2, \ldots, a_N \rangle$
 1: $\mathcal{R}_{\text{random}} \leftarrow$ Generate transitions with random actions
 2: $\hat{T} \leftarrow \text{LLM}(s_0, \mathcal{A}, \mathcal{R}_{\text{random}})$                                                ▷ Initialize transition model
 3: $\mathcal{R}_p \leftarrow \emptyset$, $\mathcal{R}_a \leftarrow \emptyset$                                               ▷ Initialize replay buffers
 4: $\Pi_H \leftarrow \text{Fast-Downward}(D, \mathcal{P})$                                  ▷ Obtain high-level plan
 5: **for** each grounded operator $\underline{\omega}_k \in \Pi_H$ **do**                              ▷ Low-level planning loop
 6:      $\pi_k \leftarrow \text{BFS}(s_0, \hat{T}, \underline{\omega}_k)$                      ▷ Find low-level actions satisfying EFF$(\underline{\omega}_k)$
 7:      Store $(s, a, s', \underline{\omega}_k)$ for all transitions in $\pi_k$ in $\mathcal{R}_p$
 8:      $\pi \leftarrow \pi \cup \pi_k$                                                ▷ Append low-level subplan
 9:      **for** each $a \in \pi_k$ **do**                                  ▷ Execute actions in environment
10:          Store $(s, a, s', \underline{\omega}_k)$ in $\mathcal{R}_a$
11:      **end for**
12: **end for**
13: **if** EFF$(\underline{\omega}_{K-1})$ holds in $s'$ after executing $\pi$ **then**          ▷ Check if final goal conditions are met
14:      **return** $\pi$
15: **else**
16:      **if** $\mathcal{R}_p \neq \mathcal{R}_a$ **then**                                 ▷ Check for mismatches
17:          $\hat{T} \leftarrow \text{LLM}(\hat{T}, \mathcal{R}_p, \mathcal{R}_a)$                        ▷ Refine transition model
18:      **else**
19:          $\Pi_E \leftarrow \text{Exploration}(s_0, D)$         ▷ Generate new high-level plan, see Section 3.2
20:          **goto** Line 5, exceute $\Pi_E$ once, then re-execute $\Pi_H$      ▷ $\Pi_E$ will be used to refine $\hat{T}$
21:      **end if**
22: **end if**

---

Formally, to refine $\hat{T}$, TheoryCoder carries out an exploratory goal, a High-Level Action (HLA) denoted by $\omega$ (suppressing the dependence on $\pi$ for conciseness), where the grounded version of this HLA is $\underline{\omega}$. We also assume we have access to the abstract representation of $s$. Simulating $\underline{\omega}$ using the current $\hat{T}$ will generate the new predicted dataset:

$$D_p = D \cup \{(s, a, r, s', \underline{\omega})\}, \tag{1}$$

where $D$ represents the replay buffer. Here, the outcome of executing $\underline{\omega}$ is an updated abstract state $\phi(s')$ that satisfies the positive effects EFF$^+$ of $\underline{\omega}$:

$$\phi(s') = \phi(s) \cup \text{EFF}^+. \tag{2}$$

Therefore, we get:

$$D_p = D \cup \{(s, a, r, s', \underline{\omega}) : \phi(s') = \phi(s) \cup \text{EFF}^+\}. \tag{3}$$

When the agent carries out $\underline{\omega}$ in the environment, it obtains the actual dataset of transitions $D_a$.

Finally, we can prompt the LLM to refine $\hat{T}$ by showing the simulated (predicted) transitions $D_p$ and the actual transitions from carrying out the actions in the environment $D_a$:

$$\hat{T}_{\text{refined}} = \text{LLM}(\hat{T}, D_p, D_a). \tag{4}$$

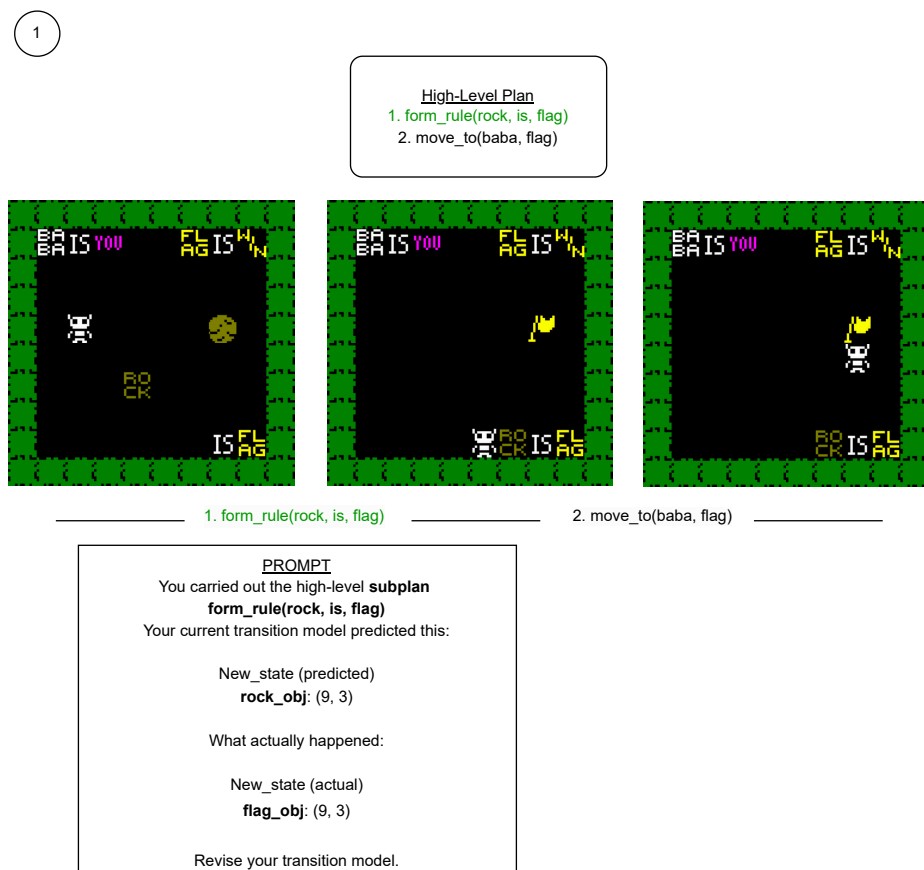

Figure 4: **Case 1. High-level plan trajectory for revision**. TheoryCoder analyzes the transitions for each subplan and finds that during the first subplan there is an error in the transition function. A prompt including this mismatch is sent to the LLM asking it to refine the transition function.

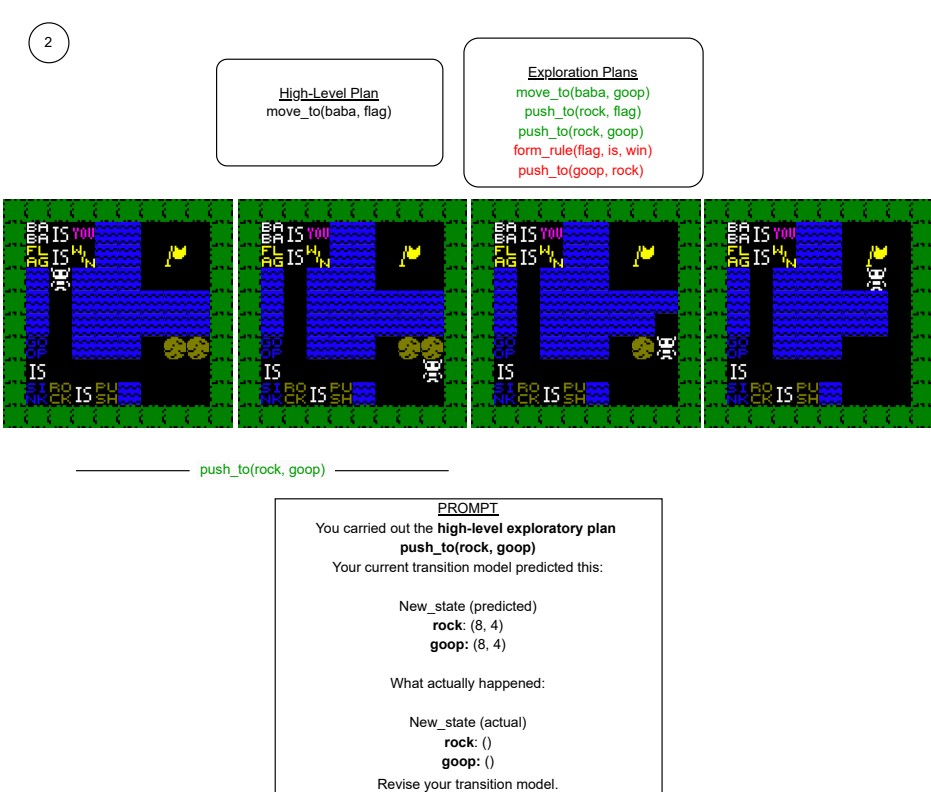

Figure 5: **Case 2. Exploratory plan**. Exploratory plans are enumerated based on those grounded operators whose preconditions are satisfied. For example, the plans in red are not viable exploratory plans since the precondition rule_formable is False for form_rule(flag, is, win). The rule cannot be formed since it's already formed. This exploratory plan is converted to a low-level action sequence to get the transitions for the replay buffer.

### 3.5 Summary

TheoryCoder is an instantiation of the TBRL framework that synthesizes Python-based world models using an LLM, rather than relying on VGDL-based theory induction as in previous work. It refines its world model through an iterative interaction loop:

- **High-level Planning**: TheoryCoder generates a high-level abstract plan using PDDL abstractions.

- **Low-level Planning**: The planner executes each subplan, translating it into a low-level action sequence.

- **Exploration and Refinement**: If the plan fails, exploratory abstract subplans are executed to collect errors in state transitions, which are used by the LLM to guide world model refinement.

## 4 Experiments

In addition to showing that TheoryCoder can handle domains not expressible in VGDL, our experiments seek to answer these questions: (1) How much improvement in sample efficiency is seen by using TheoryCoder's theory-based modeling approach compared to using an LLM directly for planning? (2) Do LLM-based agents with explicit, code-based world models learn and plan more successfully than LLM-based agents that only use their internal, implicit world models? (3) How much computational efficiency is gained by leveraging bilevel planning and abstractions compared to ablations that only plan at the level of raw states and actions? (4) How does TheoryCoder's performance on VGDL tasks compare to EMPA's? Does it solve the subset of games that EMPA failed on?

### 4.1 Evaluation and Metrics

TheoryCoder uses GPT-4o (OpenAI, 2024a) to generate and refine programs. We set the temperature to 0.7 and all other hyperparameters are set to default in the API call. The LLM-only baseline is done with GPT-o1 (OpenAI, 2024b), a more powerful model than GPT-4o, with all hyperparameters set to their default values.

We use GPT-o1 as an LLM-only agent baseline which outputs plans directly, given a text-based state representation. We use GPT-o1 rather than GPT-4o because it was specifically designed to handle more difficult reasoning and planning tasks. We focus on text-based LLM-only agents as opposed to including vision-based since vision-based agents have shown worse performance (Cloos et al., 2024).

To measure sample efficiency, we record the number of API calls each method makes. Each method is given up to 5 retries after the initial attempt per level, resulting in a maximum of 6 API calls per level. For each retry, the LLM-only agent is given the state transitions resulting from executing its previous plans. This provides it with feedback data that it can reason with to correct its plan. Note that TheoryCoder differs in that the LLM is only used to refine the python world model given state transitions, and does not directly revise plans given feedback. In our prompts, we also give a natural language description of the domain to both the LLM-only agent and TheoryCoder to provide it with context. Evaluating each method using the number of API calls allows us to quantify how effectively each method utilizes its queries to solve tasks (addresses Question 1).

Task success rate is measured as the percentage of levels successfully completed within the allotted retries. If a method fails to solve a level after five retries, it is considered a failure. This metric evaluates how effectively the model uses its world model for planning (addresses Question 2).

We measure computational efficiency as the time it takes to output a plan with and without the abstract sub-plans (addresses Question 3).

### 4.2 Environments and Tasks

We evaluate TheoryCoder in deterministic grid-style 2D games that test different aspects of planning and reasoning. We use games suitable to test LLM-style agents that have text-based state representations

available including Baba is You (Oy, 2019) and BabyAI (Chevalier-Boisvert et al., 2019). We use the KekeCompetition (Charity & Togelius, 2022) version of Baba is You. Furthermore, we evaluate our system on a set of VGDL-based games that were used to evaluate EMPA, particularly ones it struggled in.

We choose to focus our computational efficiency analysis on VGDL games, because for BabyAI and the KekeCompetition the grid-sizes are fairly small and the runtimes did not differ significantly with and without abstractions.

For each domain, we design a minimal set of operators and predicates that capture the key concepts that a human might realistically come into the game with. We find that good abstractions can be key to generating efficient plans, as we discuss below. But we emphasize that the abstractions on their own do not generate success. Like humans, TheoryCoder must learn how these concepts ground out in the particulars of each domain. For example, in Baba is You, simply possessing the concept of forming or breaking rules does not tell you that a rule of the form "X is Y" can result in transmutation, where all objects of type X are converted into objects of type Y (see results below).

### VGDL: Sokoban and Push Boulders 1

Sokoban is a game where the agent needs to push all boxes on the map into a hole. Some challenges include getting boxes stuck in corners, making it impossible to reach a win condition.

Push Boulders 1 is a task where the agent needs to navigate a grid world that has different types of poisons and boxes. Certain boxes can be pushed into certain poisons to dissolve them, ultimately allowing traversal through compact mazes.

We choose the VGDL games Sokoban and Push Boulders 1 since they are both domains that EMPA performed poorly on (Tsividis et al., 2021). Additionally, both these domains serve as good testbeds for navigation and spatial reasoning, and Push Boulders 1 tests long-term planning and causal reasoning.

### BabyAI and Minigrid

BabyAI (Chevalier-Boisvert et al., 2019) is an instruction-following domain where an agent can pick up objects and unlock color-coded doors with keys. Tasks include instructions commanding the agent to pick up objects, placing objects next to each other, and unlocking colored doors in a certain sequence. This domain mainly tests navigation and spatial reasoning. BabyAI is built on top of Minigrid (Chevalier-Boisvert et al., 2018; 2023), which share the same environment and objects.

### KekeCompetiton

Baba is You is a puzzle game where an agent can push around words to form and break rules that dictate the mechanics of the environment and how objects behave. It involves deducing environment dynamics, long-term planning and causal reasoning. We choose the KekeCompetition (Charity & Togelius, 2022) open-source version of Baba is You for our reported baseline comparisons. While another open-source version of the game exists called Baba is AI (Cloos et al., 2024), we focus on KekeCompetition, because it has more rules and serves as a more difficult evaluation. For completeness, however, **we also evaluated TheoryCoder on Baba is AI and found that it was able to solve all levels** using the world model it synthesized for the KekeCompetition.

## 4.3 Results

The results for Push Boulders 1 and Sokoban, presented in Table 1 and Table 2, illustrate how the LLM-only agent GPT-o1 struggles with long-horizon planning but TheoryCoder does not. For example, in Push Boulders 1, GPT-o1 struggled with problems that involved dissolving poisons in a certain order. In contrast, TheoryCoder is able to leverage a less powerful LLM, GPT-4o, and build an accurate world model starting from the first level. Since the world model captures most of the core mechanics, it is able to reuse this model across multiple levels. The 0 for an API call denotes that TheoryCoder did not need to call an LLM. Rather it was able to simulate the correct plan and then execute it. For example, the agent first uses 1 API call to learn that boxes can be pushed in Sokoban and packed into the holes. In the subsequent levels, the agent

| Level | GPT-o1 | TheoryCoder |
|---|---|---|
| 1 | 1 | 1 |
| 2 | 6 (failed) | 0 |
| 3 | 6 (failed) | 0 |
| 4 | 6 (failed) | 0 |
| 5 | 6 (failed) | 0 |
| **Total API Calls** | 25 | 1 |

Table 1: Sokoban API usage

| Level | GPT-o1 | TheoryCoder |
|---|---|---|
| 1 | 1 | 1 |
| 2 | 6 (failed) | 0 |
| 3 | 6 (failed) | 0 |
| 4 | 6 (failed) | 0 |
| **Total API Calls** | 19 | 1 |

Table 2: Push Boulders 1 API usage

| Agent | Steps Taken | Success Rate |
|---|---|---|
| TheoryCoder | 197 | 100% |
| EMPA | 378 | 100% |
| GPT-o1 | 9 | 20% |

Sokoban

| Agent | Steps Taken | Success Rate |
|---|---|---|
| TheoryCoder | 236 | 100% |
| EMPA | 62 | 50% |
| GPT-o1 | 26 | 25% |

Push Boulders 1

Table 3: Comparison of agent accuracy and environment steps.

can reuse this knoweldge without having to learn any new concepts and this is represented by the 0 API calls.

Table 3 shows that TheoryCoder is able to outperform EMPA by beating all Sokoban levels in half the steps. For Push Boulders 1, TheoryCoder is also able to beat the levels that EMPA failed at, given a budget of at most 500 environment steps. Figure 6 also highlights TheoryCoder's sample efficiency improvement over EMPA. Tsividis et al. (2021) introduced the learning efficiency metric $k$, which is defined as

$$k = \frac{\text{levels completed}}{\text{levels in game}} \times \frac{\text{levels completed}}{\text{steps to completion}}$$

The learning efficiency metric aims to capture both the speed and depth of learning, in that it measures how many levels of a game the agent wins and how quickly. Figure 6 shows that TheoryCoder's learning effiency is far greater than both EMPA and GPT-o1.

In terms of computational efficiency, we are able to significantly improve planning time in both VGDL games by using bilevel planning, presented in Table 4.We also compare the planning time to a pure planning approach, which encodes the entire domain into PDDL and uses Fast Downward to solve it. The PDDL files are manually designed without learning or revision. We find our approach to be much faster than this pure planning approach.

| Level | Bilevel | Ablated | FD |
|---|---|---|---|
| 1 | 0.03s | 0.07s | 0.88s |
| 2 | 0.24s | 0.76s | 1.00s |
| 3 | 0.13s | 2.61s | 5.15s |
| 4 | 0.40s | 149.70s | 13.64s |
| 5 | 24.58s | TIMEOUT | TIMEOUT |

| Level | Bilevel | Ablated | FD |
|---|---|---|---|
| 1 | 58.38s | 58.38s | 76.33s |
| 2 | 30.11s | 432.58s | 129.00s |
| 3 | 434.89s | TIMEOUT | TIMEOUT |
| 4 | 8.14s | 59.73s | TIMEOUT |

Table 4: Ablating bilevel planning abstractions for TheoryCoder on the VGDL games Sokoban (left) and Push Boulders 1 (right). Runtime is measured in seconds and averaged over 5 runs. In Push Boulders 1, level 1 runtime is the same for Bilevel and Ablated since abstract subplans are not necessary. The planner is given 500s to return a plan, after which it times out. Fast Downward (FD) struggles in levels that have lots of effects and preconditions that need to be satisfied.

The results for BabyAI, presented in Table 5, are also consistent with the results for the VGDL games above; TheoryCoder is extremely efficient compared to the LLM-only baseline. The results also once again illustrate

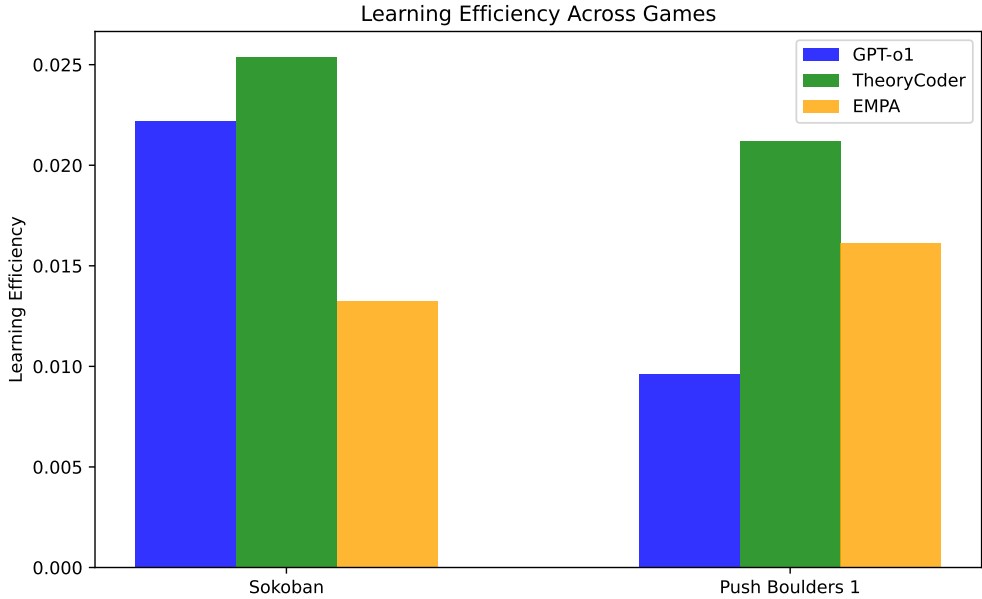

Figure 6: Learning efficiency compared across VGDL games.

a strong advantage of our approach: most of the key mechanics of BabyAI can be readily inferred in the first level, and thus TheoryCoder does not need to make very many additional API calls in subsequent levels in this case, unlike the LLM-only baseline.

| Level | GPT-o1 | TheoryCoder |
|---|---|---|
| 1 | 1 | 1 |
| 2 | 1 | 0 |
| 3 | 1 | 0 |
| 4 | 1 | 0 |
| 5 | 2 | 0 |
| 6 | 1 | 0 |
| 7 | 1 | 0 |
| 8 | 6 (failed) | 1 |
| 9 | 2 | 0 |
| 10 | 3 | 0 |
| 11 | 6 (failed) | 0 |
| 12 | 6 (failed) | 0 |
| 13 | 1 | 0 |
| 14 | 6 (failed) | 0 |
| 15 | 3 | 0 |
| 16 | 6 (failed) | 0 |
| 17 | 2 | 0 |
| **Total API Calls** | 49 | 2 |

Table 5: Comparison of GPT-o1 and TheoryCoder API usage across BabyAI levels

Additionally, we compare TheoryCoder with WorldCoder (Tang et al., 2024) on the Minigrid environments used in their paper. We train our agent using the same curriculum but find we are able to achieve far greater efficiency 7. We use a maximum depth of 30 for their MCTS planner as well as a max LLM synthesis request budget of 50. When, 50 API calls are reached and a level is not solved, then it is considered to be failed.

We found the KekeCompetition to be the most conceptually challenging domain for the LLM-only agent where it failed many of the levels, making it especially informative for Question 2. In contrast, TheoryCoder initialized a world model that captured the core mechanics of pushing around words, which naturally capture breaking and forming rules. Having captured these core mechanics, this world model did not need to be revised until Level 5.

Additionally, for this domain, we tested two levels of prompting for GPT-o1: minimal, (providing the state and description of the domain) and heavy (providing the high-level plan that would win the level as well as the ground truth transition model). Even with heavy prompting, GPT-o1 was not able to improve its results much, as shown in Table 6.

| Level | GPT o1 | | TheoryCoder |
|---|---|---|---|
| | minimal | heavy | |
| 1 | 1 | 1 | 1 |
| 2 | 2 | 2 | 0 |
| 3 | 6 (failed) | 6 (failed) | 0 |
| 4 | 1 | 1 | 0 |
| 5 | 6 (failed) | 3 | 1 |
| 6 | 1 | 2 | 0 |
| 7 | 1 | 1 | 0 |
| 8 | 1 | 1 | 0 |
| 9 | 1 | 1 | 0 |
| 10 | 6 (failed) | 6 (failed) | 1 |
| 11 | 6 (failed) | 6 (failed) | 4 |
| 12 | 1 | 1 | 0 |
| 13 | 6 (failed) | 6 (failed) | 0 |
| **Total API Calls** | 39 | 37 | 7 |

Table 6: Comparison of GPT o1 and TheoryCoder API usage across KekeCompetition levels

| Agent | Total Tokens | Success Rate |
|---|---|---|
| TheoryCoder | 69,402 | 100% |
| WorldCoder | 388,861 | 83% |

Table 7: Comparison to WorldCoder on Baby AI domain. The same curriculum that WorldCoder was trained on is used for TheoryCoder. 50 API calls are allowed per level, after which it is considered unsuccessful.

## 4.4 Role of Abstraction Design

In Sokoban, the goal state is to push all boxes into the holes. However, attempting to find a low-level action plan for this goal state would be very costly. We are able to significantly improve planning time by using a simple abstraction *push_to_hole*, that the agent should aim to push one box into a hole at a time. Using this abstraction we are able to divide the planning problem and find a low-level plan to store each of the boxes. Then, we concatenate the plans at the end. This one abstraction much improves computationally efficiency, as shown in Fig. 4. However, one crucial effect must be added to this operator, the *boxes_stuck* predicate. Without this predicate, the agent could successfully store a box but at the same time push another box into a corner and make it stuck, resulting in a loss. Thus, we include the predicate *boxes_stuck* to make sure that each time the agent pushes a box into a hole no other boxes are stuck.

The abstractions for Push Boulders 1 can be seen in Fig. 8b. We emulate a strategy where humans tend to segment different regions of mazes. Our abstraction *move_between_bottlenecks* segments regions of the maze. For example, in Fig. 8a, it can be seen that the maze has three regions. The dark blue avatar begins in the top-left region and the pink goal square in the top-right region surrounded by yellow poison. One abstraction for this environment is to divide it into rooms or regions based on the bottlenecks, the narrow

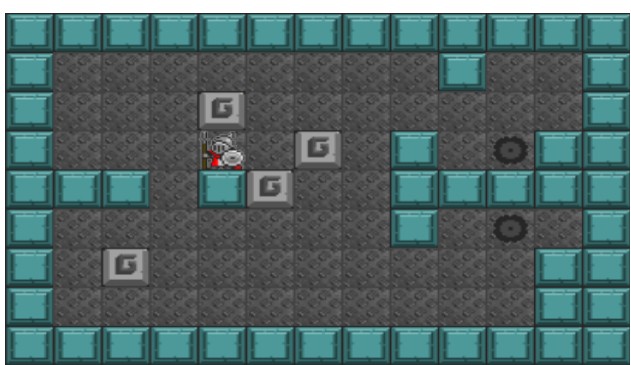

(a) Level of Sokoban

```
(:predicates
    ; Box not in hole yet
    (unstored_box ?box - box)
    ; Checks if any box in the state is stuck
    (boxes_stuck)
)

(:action push_to_hole
    :parameters (?box - box)
    :precondition (unstored_box ?box)
    :effect (and (not (unstored_box ?box)) (not (boxes_stuck)))
)
```

(b) Sokoban Abstractions

Figure 7: (a) Sokoban environment and (b) its corresponding PDDL abstractions.

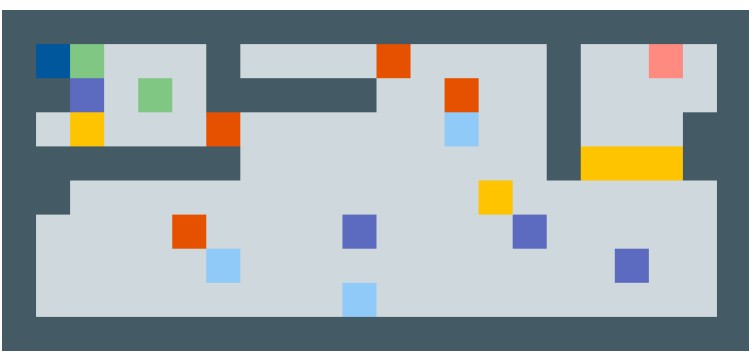

(a) Level of Push Boulders 1

```
(:predicates
    (at ?place) ; Agent is at a specific location
    (connection ?from ?to) ; Path exists between two locations
)

(:action move_between_bottleneck
    :parameters (?from - object ?to - object)
    :precondition (and (at ?from) (connection ?from ?to))
    :effect (and (at ?to))
)
```

(b) Push Boulders 1 Abstractions

Figure 8: The push boulders level can be seen as having three regions. The small top-left region containing the dark blue avatar square, the open middle region and then top-right region with the pink square. Each of these regions can be seen as bottlenecks which the agent will move between.

parts of the map where walls start to close in. This abstraction allows TheoryCoder's planner to significantly improve its planning runtime since it breaks up search by finding three separate plans, for each of the regions, then finally concatenating it into one overall plan. The computational efficiency gain is substantial, as seen in Table 4.

In Table 7, we again see the usefulness of our built-in abstractions, which guide exploration and learning. Our agent solves the levels with significantly less cost compared to WorldCoder (Tang et al., 2024), another program-based world model learning agent.

For both VGDL domains, the mechanics that need to be learned are simple; TheoryCoder is able to synthesize a correct low-level model in the initial API call during each of the first levels. As such, the abstractions are not critical for sample efficiency since no world model revision was needed. The VGDL domains mainly demonstrate the computational efficiency achieved by using bilevel planning, as the VGDL domains have the largest state spaces (particularly Push Boulders 1, which requires navigating long mazes).

For BabyAI, the simple abstraction "put_next_to" was helpful for delegating spatial reasoning that involved adjacent objects. On the other hand, the LLM-only method commonly made errors in this type of reasoning and had to correct itself. The "drop" also helped TheoryCoder by making explicit the key steps required to clear a doorway. In particular, the effect "not_near_door" helped TheoryCoder realize that when a door is unlocked, the key should not be dropped in doorway, or else the passage to the next room would be blocked. The LLM-only struggled to realize this and blocked the doorway.

In Level 5 of the KekeCompetition 2, the rule "rock is flag" has to be formed, converting the rock into a flag. The abstraction $form\_rule$ was likely useful in guiding the LLM's reasoning here, as it captured this latent state (whether the flag appears on the map).

TheoryCoder required relatively more rounds of revision to learn the world model for Level 11, depicted in Fig. 11a. The goal of this level is to push rocks into the "goop", which removes them both from the state, clearing a path. TheoryCoder tackled this level by carrying out an abstract exploratory sub-plan of pushing rocks into goop. Then GPT-4o changed the world model to include the notion of rocks being removed when encountering goop, but failed to specify that the goop should also be removed. A second API call was used to revise the model to remove goop when baba overlaps with it. In the third API call, GPT-4o made an irrelevant change by marking the flag as an object that can be overlapped. Finally, in the fourth API call, GPT-4o correctly revised TheoryCoder's world model to include the line of code that goop is removed along with rock when they overlap.

### 4.5 Overall Sample Efficiency and Progress Across Domains

As shown in Table 8, TheoryCoder uses significantly fewer API calls than GPT-o1. TheoryCoder can reuse its world model to simulate plans in levels with similar mechanics without needing to use an API call to request a plan. Fewer API calls and tokens are important metrics since using these powerful close-source LLMs can incur financial cost rapidly, even in these relatively smaller domains.

We find that TheoryCoder's world model also results in more progress across the games, as shown in Fig. 12. This follows from works such as Gupta & Kembhavi (2023) which also found that structured program generation can be a useful way to improve reasoning (in their case visual reasoning). This is likely due to the structured approaches helping guide the LLM generation by constraining its output.

| Game | GPT-o1 | TheoryCoder |
|---|---|---|
| KekeCompetition | 39 | 7 |
| BabyAI | 49 | 2 |
| Sokoban | 25 | 1 |
| Push Boulders 1 | 19 | 1 |

Table 8: Comparison of TheoryCoder and GPT-o1 (API calls required).

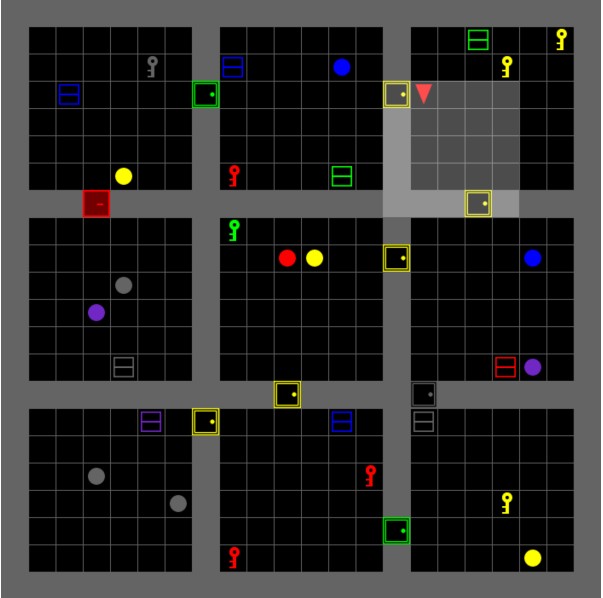

(a) A level from BabyAI

```
(:predicates
    (carrying ?obj - object)
    (next_to ?obj1 - object ?obj2 - object)
    (unlocks ?key - object ?door - door)
    (blocking ?obj - object ?door - door)
    (clear ?door - door)
    (inventory_full)
    (agent_moved_away ?door - door)
    (not_near_door ?obj)
)

(:action drop
        :parameters (?obj - object)
        :precondition (and (carrying ?obj) (inventory_full))
        :effect (and (not (carrying ?obj)) (not (inventory_full)) (not_near_door
            ?obj))
    )

(:action unblock
    :parameters (?door - door ?obj - object)
    :precondition (and (blocking ?obj ?door) (not (clear ?door)) (not (carrying
        ?obj)) (not (inventory_full)))
    :effect (and (agent_moved_away ?door) (not (blocking ?obj ?door)) (clear ?door)
        (carrying ?obj) (inventory_full))
)

(:action put_next_to
    :parameters (?item - object ?adjacent - object)
    :precondition (not (next_to ?item ?adjacent))
    :effect (next_to ?item ?adjacent)
)
```

(b) BabyAI Abstractions

Figure 9: (a) BabyAI environment, and (b) its corresponding PDDL abstractions.

**World Model Revision**

```
ERRORS FROM WORLD MODEL for ABSTRACT PLAN form_rule rock_word is_word flag_word:

{Initial State}

{State after ABSTRACT PLAN}

Your prediction errors:
"overlappables": predicted: []
"overlappables": actual: ['flag_obj']

"rules_formed": Missing: ['rock_word is_word flag_word']

Key mismatch: "rock_obj" is missing, but "flag_obj" has the same coordinates.

Key mismatch: "flag_obj" is missing, but "rock_obj" has the same coordinates.
```

```python
def manage_rule_changes(state):
    # Check if 'rock_word is_word flag_word' rule is formed
    if 'rock_word is_word flag_word' in state['rules_formed']:
        # Convert rock_obj to flag_obj and manage overlappables
        if 'flag_obj' not in state:
            state['flag_obj'] = []
        state['flag_obj'].extend(state.get('rock_obj', []))
        state.pop('rock_obj', None)

        # Mark flag_obj as overlappable
        state['overlappables'].append('flag_obj')
```

Figure 10: TheoryCoder revises its world model by prompting GPT-4o with a replay buffer delineated by an abstract plan label, as opposed to using a large action-by-action replay buffer. This helps during world model revision since the abstract plans help the LLM see the error at a high-level.

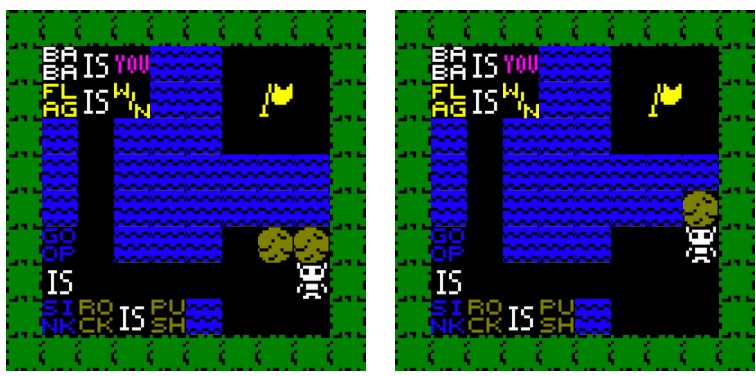

(a) Level 11: Pushing rocks into goop clears path

```
(:predicates
    (rule_formed ?word1 - word ?word2 - word ?word3 - word)
    (rule_formable ?word1 - word ?word2 - word ?word3 - word)
    (rule_breakable ?word1 - word ?word2 - word ?word3 - word)
)

(:action form_rule
    :parameters (?word1 - word ?word2 - word ?word3 - word)
    :precondition (and (not (rule_formed ?word1 ?word2 ?word3)) (rule_formable
        ?word1 ?word2 ?word3))
    :effect (rule_formed ?word1 ?word2 ?word3)
)

(:action break_rule
    :parameters (?word1 - word ?word2 - word ?word3 - word)
    :precondition (and (rule_formed ?word1 ?word2 ?word3) (rule_breakable ?word1
        ?word2 ?word3))
    :effect (not (rule_formed ?word1 ?word2 ?word3))
)
```

(b) KekeCompetition Abstractions

Figure 11: KekeCompetition and its abstractions

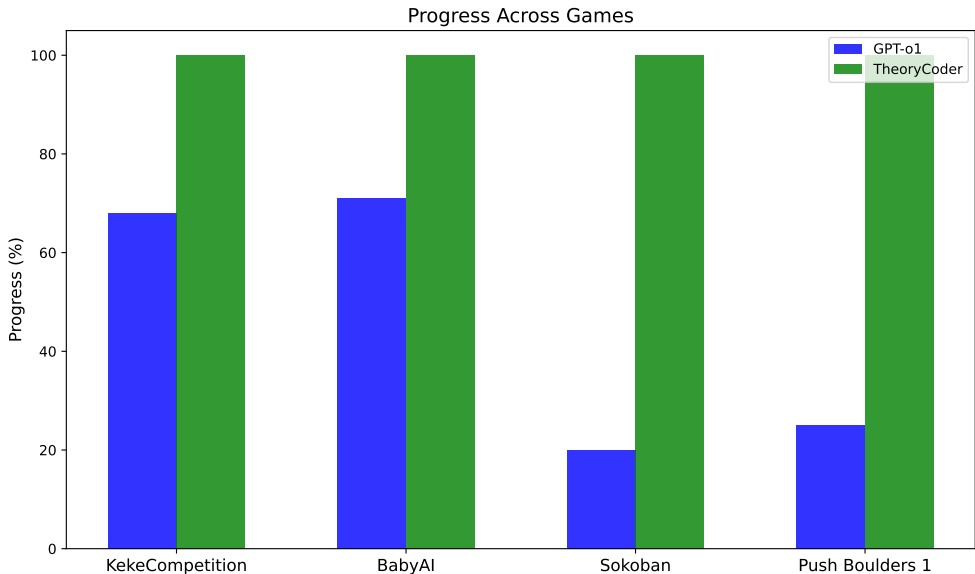

Figure 12: Progress comparison between GPT-o1 and TheoryCoder across different games.

We found that the abstract PDDL subplans were helpful during TheoryCoder's refinement. By including the subplans, the prompt could be directed towards refining one particular mechanic at a time. On the other hand, we found that without the abstract subplans, the LLM could overlook certain errors during refinement, which would result in it needing another API call to correct that error. As shown in Fig. 2, the replay buffer labels the section of the state transitions that deal with certain abstract plans. This allows the LLM to see the consequences of its low-level action sequence and use the high-level semantic knowledge to correct the world model more precisely.

Finally, we found that the sub-plans significantly improved TheoryCoder's planning times (Table 4). Since each abstract plan encodes an intermediate goal state for the BFS planner, it allows breaking the planning problem into more manageable stages, similar to divide-and-conquer strategies. In some cases, hours of planning during runtime were reduced to minutes.

We also speculate that this method of building a world model could lead to shorter planning times compared to making API calls to an LLM-only reasoning model, particularly in small to medium-sized worlds. However, a more sophisticated planner would likely be needed. For larger or more complex environments, the computational cost of symbolic planning could grow, and the efficiency trade-offs between program-based planning and LLM-based reasoning remain an open question. A hybrid approach using the LLM as an implicit heuristic may work well.

One interesting property of code-based world models are the potential for compositionality and transfer. Once an underlying concept is represented as generalizable code, this code can be used to solve any variation of the task. For example, code can enable adapting seamlessly to arbitrary changes in the goal object and its color—whether the target is a key, a ball, or a flag. An interesting future line of work would be to study whether this could transfer across visually distinct games with the same underlying concept.

### 4.6 Behavior Comparison

We compare the behavior of GPT-o1 and TheoryCoder, shown in Fig. 13 and find that TheoryCoder's exploration patterns are more object-oriented following TBRL patterns. In level 11, having already learned that goop is dangerous, TheoryCoder explores by pushing a rock into goop to see how they interact. We give GPT-o1 the high level plan that can allow it to win. However, we find that GPT-o1 struggles with spatial reasoning and repeatedly falls in the goop. In level 13, TheoryCoder is able to solve the level in its first

attempt. GPT-o1 initially attempts to follow the high level plan of forming "Keke is You" but fails to do so. Following this, GPT-o1 disregards the given high level plan and repeatedly attempts to break the "Wall is Stop" rule. We find that GPT-o1 tends to commit to one line of reasoning.

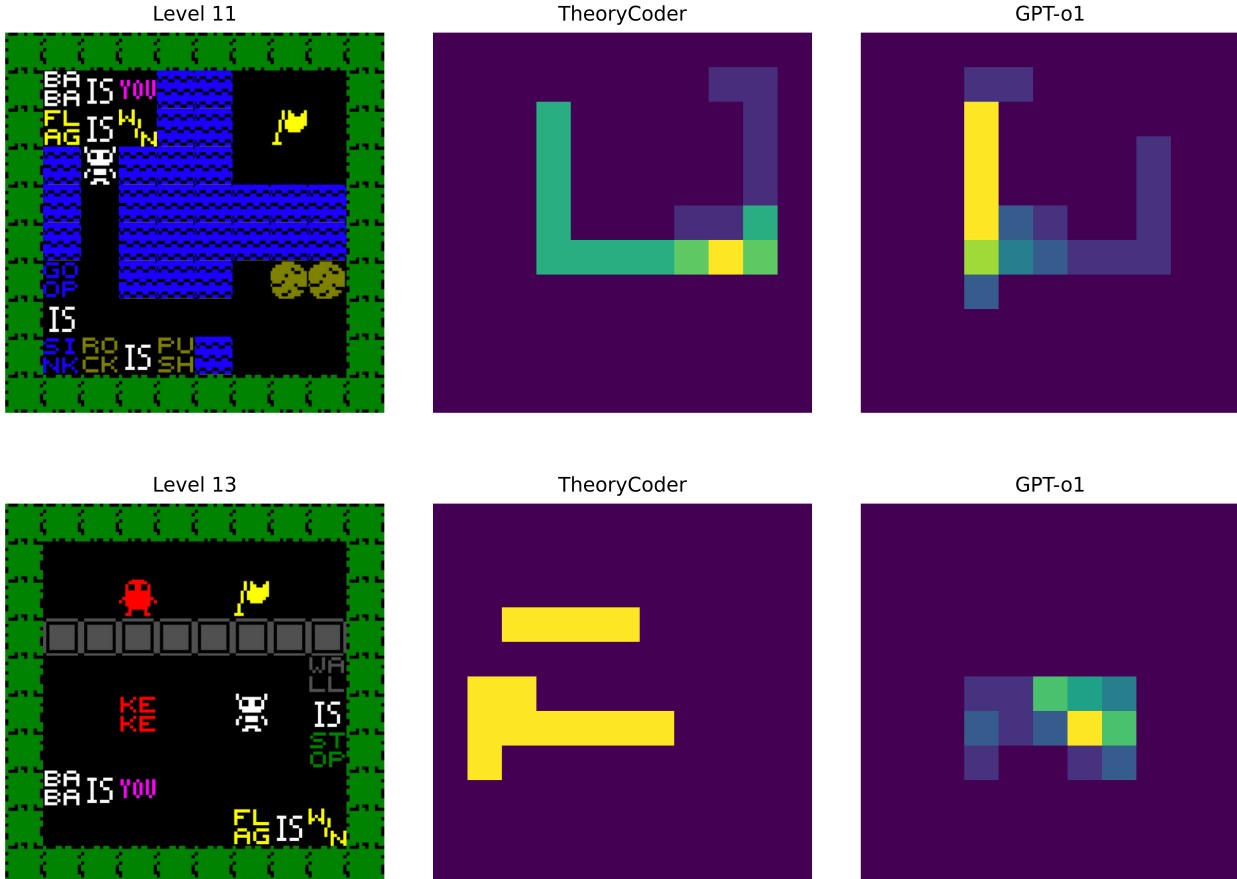

Figure 13: Game-play Heatmap Comparison between TheoryCoder and GPT-o1

## 4.7 Analysis of Robustness and Failure

We found that although TheoryCoder could consistently generate world models when run multiple times in the same environment, in some of these runs it would get stuck. This would happen at the beginning of the game, which would make it difficult for our agent to recover. An example of this is in the KekeCompetition where if the agent was unable to come up with the correct code that represents pushable objects can push one another, then it would fail later on in the levels and not be able to start learning the changing rule dynamics, unless the run was restarted.

Furthermore, to increase robustness of model synthesis, we included some code syntax instructions in the prompt and encouraged it to be more general.

To evaluate the limitations of TheoryCoder, we tested the system on two KekeCompetition levels outside of the standard demo set. In the first level on the left in Fig. 14, TheoryCoder successfully learned key subplans, such as the interaction that requires pushing a heart into a skull to escape the enclosure. However, TheoryCoder failed to complete the level even after 25 API calls. This is because the final step of the task requires sacrificing two of the controlled characters by moving them into goop, which will destroy the goop and clear the path for the remaining controlled character to reach the flag. This is a strategy that contradicts the agent's learned rule that goop causes death and should be avoided. This highlights a limitation in TheoryCoder's revision strategy: the world model may overgeneralize from early transitions.

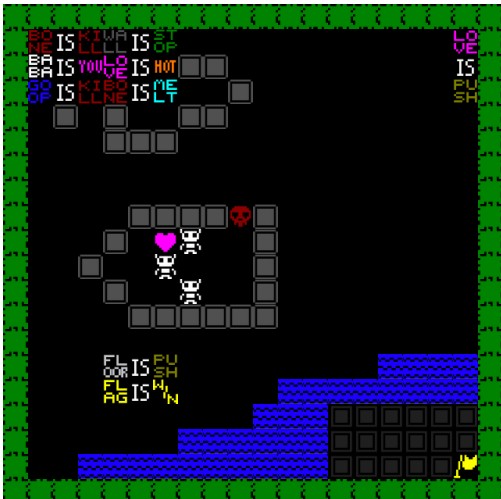 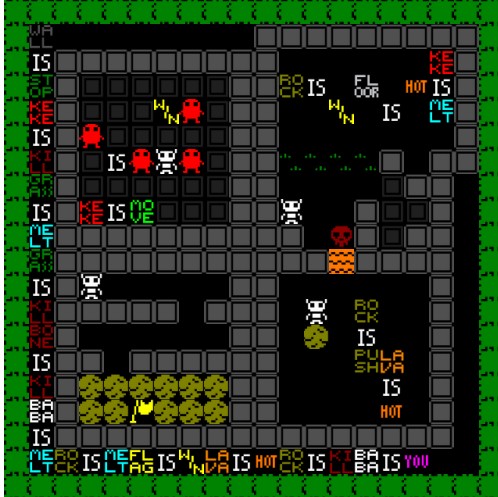

Figure 14: TheoryCoder failure mode analysis. The level on the left requires the agent to sacrifice two of the controlled Baba characters to win. The level on the right also requires sacrificing one Baba character, while keeping one constantly alive.

In this case, it overgeneralizes the rule (e.g., "goop causes death") and fails to consider that it may be conditionally useful, such as in this level that requires dying in order to win. The second level on the right in Fig. 14 is another level which shares the same principle of having to sacrifice one of the Baba characters. In this level, the agent must constantly keep the bottom left Baba alive by preventing its movement into the rocks. At the same time, it must move the Baba in the top right area into the skull to clear part of the path. TheoryCoder was again unable to solve this level due to its world model overgeneralizing.

Future work can address these issues by extending TheoryCoder with a set of world models and cycling between them when one repeatedly fails. Additionally, the agent can be prompted to think more flexibly and to challenge its current assumptions. This would be important for tackling more naturalistic tasks that require challenging assumptions which may have always held true until now.

### 4.8 Excluding Training Data Contamination

TheoryCoder uses GPT-4o, which currently has a training data knowledge cutoff of October 2023 (OpenAI, 2024a). It may be possible that the codebases for our environments were included in this training data. To test whether our results hold regardless of this, we run an experiment on a newly designed custom game that could not have been seen by GPT-4o. In our custom designed game, the objective is to cluster different colored boxes together, often while having to navigate some maze. In some levels there are cherries which are lethal to the agent and must be avoided, going against the prior that cherries are positive items.

We find that even in this environment, our agent follows the same trend of being able to learn efficiently with few interactions as shown in Fig. 15.

## 5 Conclusion

In this work, we introduced TheoryCoder, a novel instantiation of TBRL that integrates LLM-based program synthesis with bilevel planning. Our approach enables the generation of more expressive theories and facilitates more efficient planning, extending the applicability of TBRL beyond traditional Atari-style domains.

Overall, our findings support the hypothesis that stratified learning strategies applied to hierarchical abstractions can improve planning and learning efficiency (Sutton et al., 1999) in non-trivial problem domains like Baba is You. While our method relies on highly sophisticated search methods to quickly synthesize

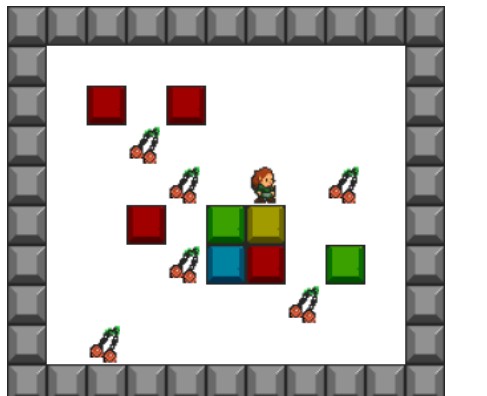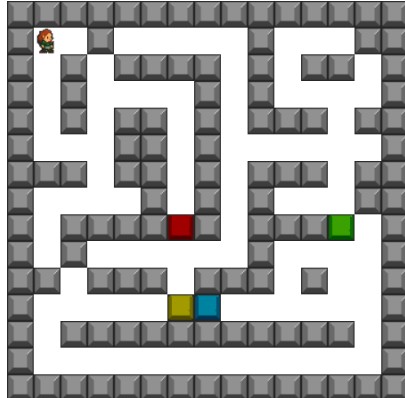

| Level | API Calls | Total Tokens |
|-------|-----------|--------------|
| 1 | 1 | 9645 |
| 2 | 0 | 0 |
| 3 | 1 | 22134 |
| 4 | 2 | 12816 |
| 5 | 0 | 0 |
| 6 | 0 | 0 |
| 7 | 0 | 0 |
| 8 | 0 | 0 |
| 9 | 0 | 0 |
| 10 | 0 | 0 |

Figure 15: TheoryCoder API Calls and Token Usage for newly designed game.

programs in open-ended languages, we find that LLMs are at least one kind of system that can be used successfully in this way.

Our findings have broad implications for the science of learning, both for cognitive and AI scientists. The high sample efficiency exhibited by TheoryCoder suggests new fruitful directions to explore regarding how abstractions can be leveraged for fast and adaptive learning.

Through evaluations on complex environments including Baba Is You, BabyAI, and VGDL-based puzzle tasks such as Sokoban, TheoryCoder significantly outperformed LLM-only approaches in both sample efficiency and planning effectiveness. Notably, TheoryCoder achieves up to a $24.5\times$ reduction in API usage, highlighting its ability to refine world models and reuse knowledge across similar levels rather than relying on costly new queries. Additionally, TheoryCoder successfully solved VGDL tasks where EMPA previously struggled, showcasing the benefits of adding stratified learning and hierarchical planning to the TBRL toolkit.

**Limitations and future directions.** While our results show that TheoryCoder is an effective alternative to LLM-only planning, there are several limitations to consider. Hand engineering useful abstractions can be tedious and poses a clear challenge for the scalability and generality of the approach. People can also differ on what their views about what is the "right" set of abstractions for any given domain; there is no single ground truth here. Our future work will tackle the more challenging problem of how to learn or construct the abstractions that get transferred in the first place. For example, it may be possible to use LLMs to formalize the problems they encounter by generating PDDL domain files. Given recent work using LLMs to construct full PDDL models, including abstract operators and predicates (Wong et al., 2024; Guan et al., 2023), we suspect this may be a fruitful direction. Another technical challenge that should be addressed in future work concerns the application of symbolic planners to dynamic, continuous domains. Future work could draw on techniques developed within the task and motion planning and related literature to connect symbolic and continuous representations. This might involve, for instance, given agents the ability to "mentally" simulate nonlinear physical dynamics using physics engines.

Future research could seek to extend TheoryCoder's capabilities in a number of ways. For instance, by exploring 3D domains, stochastic (rather than deterministic) domains, and domains that require more sophisticated exploratory mechanisms or low-level planning algorithms. Finally, it could consider multi-agent settings, exploring ways to learn useful operators and predicates by communicating with other agents through dialogue or text mining.

**Acknowledgments**

We thank Kazuki Irie and Nada Amin for thoughtful comments and discussions.

This work was supported by the Kempner Institute for the Study of Natural and Artificial Intelligence, and by the Department of Defense MURI program under ARO grant W911NF-23-1-0277.

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

# A   Appendix

Below are example prompts.

**Transition Model Initialization Prompt**

The prediction errors denote the difference between the actual replay buffer and replay buffer predicted during simulation using the transition model.

```
You are an AI agent that must come up with a transition model of the game you are playing.

A BFS low-level planner that will use your synthesized transition model to find the low-
level actions that will allow you to win levels of the game.

You are also given state transition after executing random actions that will help as well.
Note that if there is no change returned after doing that action, it means that moving was
prevented somehow such as by an obstacle.

The levels you start out with will be simpler but you will be adding on more and more as
time progresses.
So try to make the transition model general and avoid hardcoding anything from the state
dictionary keys. Feel free to infer the types of interactions that will occur in later
levels.
Do not feel like you need to build the transition model for just this replay buffer.

{DESCRIPTION OF DOMAIN}

NOTES:

Also, remember to return the state if you modified it directly or return the new_state if
you deepcopied it.

Remember to make your world model general you can use string checks like .endswith or .
beginswith instead of hardcoding a list of entities in your world model.

You can replace the keys of any state dictionary items that transform!

YOU MUST ATTEMPT TO MODEL ALL THE MECHANICS FROM THE ACTION SPACE.

{CURRENT STATE}

{ACTION SPACE}

{REPLAY BUFFER}

{PREDICTION ERRORS}

UTILS:

directions = {
    'left': [-1, 0],
    'right': [1, 0],
    'up': [0, 1],
    'down': [0, -1],
}

RESPONSE FORMAT:

- Make sure that you return the correct state for example if you made a deepcopy of the
state and modify the deep copy then return the new_state
- If you modify the state directly then return the state instead of new_state
```

```
- Try to generalize your world model. For example, do not just assume that all entities
will be the same color in all variations
- Your world model will be used in other variations where entities will be of different
colors, but the principles should remain the same
- For example, you can use string checks like .endswith instead of hardcoding a list of
entities in your world model.

'''python

# make sure to include these import statements
from copy import deepcopy
from utils import directions

def transition_model(state, action):

        Return State

'''
```

## Transition Model Revision Prompt

TheoryCoder revision prompt. The errors from world model section in the prompt is the difference between actual replay buffer and predicated replay buffer. The replay buffer is gathered for an exploratory plan denoted by a grounded operator. Each exploratory plan will have its own replay buffer.

```
You are an AI agent that must come up with a model of the game you are playing. This model
you are making of the game
will be a python program that captures the logic and mechanics of the game. You have begun
this world model, but it did not capture everything.
Below is your current world model, the action space, and the state transition that your
transition model handled wrong.
The state transition (inital state, action, next state) will be followed by a section
detailing your prediction errors.
If the prediction errors is blank it means your world model correctly modeled that
transition.

In order to craft the world model and get this state transition you explored your
environment with an EXPLORATION PLAN.
The state transitions belonging to an EXPLORATION PLAN will be written below it.
Note this exploration is a high level plan and the transitions related to it
are carrying out this high level plan by
executing actions in the {ACTION SPACE}

Pay close attention to what is involved and modify your transition model to be able to
handle this.

{DESCRIPTION OF DOMAIN}

NOTES:

Feel free to also explain your thinking outside of the markup tags, but know that I will
only use the code inside the markup tags.

The exploration plans are set up to help guide you to your overall goal.

You can replace the keys of any state dictionary items that transform!

{ACTION SPACE}
```

```
{CURRENT WORLD MODEL}

ERRORS FROM WORLD MODEL for EXPLORATORY PLAN {GROUNDED OPERATOR}:

ERRORS FROM WORLD MODEL for EXPLORATORY PLAN {GROUNDED OPERATOR}:

UTILS:

directions = {
        'left': [-1, 0],
        'right': [1, 0],
        'up': [0, 1],
        'down': [0, -1],
    }

RESPONSE FORMAT (make sure to include your code in markup tags):

- Make sure that you return the correct state for example if you made a deepcopy of the
state and modify the deep copy then return the new_state
- If you modify the state directly then return the state instead of new_state
- Try to generalize your world model. For example, do not just assume that all entities
will be the same color in all variations
- Your world model will be used in other variations where entities will be of different
colors, but the principles should remain the same
- For example, you can use string checks like .endswith or .beginswith instead of
hardcoding a list of entities in your world model.

'''Python

# make sure to include these import statements
from predicates import *
from copy import deepcopy
from utils import directions

def transition_model(state, action):

        Return State

'''
```