# OpenReview forum: "Synthesizing world models for bilevel planning"
_TMLR — Accepted by TMLR_

### Review · Reviewer_UsbX · 2025-05-28

**Summary Of Contributions:**

This paper introduces TheoryCoder, a theory-based reinforcement learning (TBRL) system that addresses key limitations of prior work, particularly EMPA (Tsividis et al., 2021). It integrates LLM-based program synthesis with bilevel planning, using Python for low-level transition models and PDDL for high-level abstractions. This replaces EMPA's restrictive VGDL-based approach with a more expressive and flexible framework. The system employs a novel bilevel planning architecture where Fast Downward handles high-level PDDL planning while breadth-first search executes low-level Python models. The learned transition model is refined with semantically labeled replay buffers. The authors demonstrate TheoryCoder's substantial improvements in efficiency and transferability over LLM-only baselines on VGDL tasks.

**Audience:**

Yes

**Claims And Evidence:**

Yes

**Requested Changes:**

* The author should consider adding some analysis of failure modes or robustness, as well as the planning/behaviors comparison between TheoryCoder and GPT-o1.

**Strengths And Weaknesses:**

__Strengths__

* TheoryCoder uses Large Language Models and a Python-based learnable transition model, which tackles known limitations of EMPA, such as restricted expressiveness. This allows much more complex world models.
* The bilevel planning architecture is well-designed. Combining an exploratory plan with abstract labels to guide LLM refinement helps the LLM focus on specific errors to refine the transition model.
* The efficiency improvements against GPT-o1 on a series of test sets are substantial and consistent across domains. The transfer learning from KekeCompetition to Baba AI is interesting.

__Weakness__

* The problem and domain files are manually designed. While acknowledged as future work, this limits practical applicability compared to methods that learn abstractions.
* All tested domains are relatively small grid worlds. The author may consider testing whether this symbolic planning framework works on larger, more complex problems that require long-term memory and planning.
* The api calls might not be a perfect efficiency metric. The token usage may serve as another metric indicating efficiency.
* The paper could be enhanced with some analysis of failure modes or robustness, as well as the planning/behaviors comparison between TheoryCoder and GPT-o1.

__Question__

* The transferability is interesting. Is it because here, between these two games, the state and game mechanism are almost the same, and only the layouts are different (which looks like not being input to the world model)?

---

> ### Author Response · Authors · 2025-06-05
> **Response to Reviewer UsbX**
>
> We sincerely thank the reviewer for their positive remark on the design of our architecture and interest in our work. We address the weaknesses and requested changes below.
>
> > The problem and domain files are manually designed. While acknowledged as future work, this limits practical applicability compared to methods that learn abstractions.
>
> We agree with this comment and aim to address this in future work.
>
> > All tested domains are relatively small grid worlds. The author may consider testing whether this symbolic planning framework works on larger, more complex problems that require long-term memory and planning.
>
> Thank you for this feedback. We want to mention that in one of our evaluated domains, BabyAI, our system is able to handle the levels that require long-term planning such as navigating through multiple rooms while finding keys, unblocking doorways and unlocking rooms. We updated Figure 9. to include an image of one of these more difficult levels that our system successfully solved. However, we do agree that future work would be interesting to test this type of symbolic approach on more complex problems especially those with long-term memory and planning.
>
> > The api calls might not be a perfect efficiency metric. The token usage may serve as another metric indicating efficiency.
>
> Thank you for this suggestion. In our new experiments for the revision, we now document token usage as well.
>
> > The transferability is interesting. Is it because here, between these two games, the state and game mechanism are almost the same, and only the layouts are different (which looks like not being input to the world model)?
>
> Yes that is correct, it is able to transfer since the state and game mechanics are the same. However, the key difference is that there are new objects in one game that are not present in the other and these new objects are crucial for solving the task. Systematically testing this with more games and deeper analysis would be an interesting direction for future work.
>
> > **(Requested changes)** The author should consider adding some analysis of failure modes or robustness, as well as the planning/behaviors comparison between TheoryCoder and GPT-o1.
>
> - We have added section 4.7 which provides an analysis of robustness and failure modes of our system on more challenging tasks, accompanied by Figure 14.
>
>   - We analyze tasks where our system has substantial room for improvement and suggest the changes that should be made.
>   - Mainly, we found that the synthesized world model can be overgeneralizable in certain tasks.
>     - For example, if the world model represents that all goop should be avoided to survive, then this world model breaks in a level where you need to purposefully die in order to win.
>   - We suggest that these types of tasks can be solved by extending TheoryCoder to build a set of world models that can be cycled through, instead of one single unified one.
>
> - We have added section 4.6 which compares the behavior of GPT-o1 and TheoryCoder along with Figure 13 that includes heatmaps of gameplay.
>
>   - We find that TheoryCoder’s exploration patterns are more object oriented, following previous TBRL systems.
>   - We also find that o1 tends to commit to one strategy and reasoning even when it repeatedly fails.
>
> To conclude, we thank the reviewer again for their valuable insights and ideas, which have helped improve our analysis.

---

### Review · Reviewer_3KZ9 · 2025-05-28

**Summary Of Contributions:**

This paper introduce TheoryCoder, a theory-based reinforcement learning algorithm that uses LLMs for synthesizing Python-based world models and integrates these with bilevel planning via PDDL. TheoryCoder improves data efficiency, generalization ability, and planning time on tasks like BabyAI, Sokoban, and Baba Is You compared to an LLM-only baseline and EMPA, an existing theory-based RL algorithm.

**Audience:**

Yes

**Broader Impact Concerns:**

No concerns.

**Claims And Evidence:**

Yes

**Requested Changes:**

1. While the paper isn't difficult to parse, readability would be improved if the authors added additional headers. As an example, the related work has two subsections that contain 6 paragraphs, and it feels a bit like a "wall of text." Adding a few bold words at the start of new ideas would help guide the reader. For instance, the last 3 paragraphs of "Model based reasoning" focus on LLMs, you could indicate these paragraphs focus on LLMs here.


1. “Second, the planning algorithms used in previous TBRL agents are not scalable to large state spaces.” Brief describe why.
1. Section 1: the meaning of "error distribution" is unclear.
1. Section 1: clearly describe what is meant by "in-context" here
1. Section 3.2: “Thus, the optimal plan is the shortest sequence of actions from s1 to sN = s∗.” nitpick: *an* optimal plan is the shortest action sequence that reaches the goal. If you include discounting, then it is *the* optimal plan.
1. Algorithm 1, line 9: The comment says "Execute actions in the environment," though we are sampling next states from the world model $\hat T$.
1. Section 4.3: “a less powerful LLM, GPT-4o” mention that 4o is less powerful than o1 in Section 4.1.

**Strengths And Weaknesses:**

## Strengths
Overall, the paper is well written and easy to follow. The authors do a good job scoping their contributions. Empirically,
1. TheoryCoder increases data efficiency (measured as the # of LLM API calls) and decreases planning time compared to a naive LLM-only baseline.
1. TheoryCoder outperforms EMPA on VGDL tasks where EMPA is known to struggle.
1. TheoryCoder generalizes to new domains

## Weaknesses
The weaknesses below highlights points the paper hwere I was a bit confused.

1. Table 3, right: Why the sudden increase in planning time at level 3? I thought that since the model was already very good at level 1, planning times would be very small for future levels -- similar to the left table.
1. Table 3, right: “In Push Boulders 1, level 1 runtime is the same for Bilevel and Ablated since abstract subplans are not necessary.”  I'm a little confused on the decision to omit these two values from the table. Why not just include the time and then mention why their values are the same?
1. It great that TheoryCoder performs very well on these tasks and outperforms the baselines, but the tasks are "too easy" for theory coder for us to understand failure modes of the method. In particular, TheoryCoder only needs to revise the model in BabyAI and KekeCompetition -- and it only revises the model once in BabyAI. It would be useful to include more tasks where TheoryCoder has substantial room for improvement. Do the authors know what such tasks might look like?
1. How does theory-based RL relate to the options literature [1]? I'm not necessarily suggesting this framework should be discussed in the related work; it would just help clarify the contributions in my case. The conclusion states "“our findings support the hypothesis that stratified learning strategies applied to hierarchical abstractions can improve planning and learning efficiency in non-trivial problem domains like Baba is You,” but isn't this claim already supported (for tasks outside of Baba Is You) by existing work on options, hierarchical RL, etc?

[1] Sutton et. al. "Between MDPs and semi-MDPs: A framework for temporal abstraction in reinforcement learning." https://people.cs.umass.edu/~barto/courses/cs687/Sutton-Precup-Singh-AIJ99.pdf

---

> ### Author Response · Authors · 2025-06-05
> **Response to Reviewer 3KZ9**
>
> We sincerely thank the reviewer for their positive remarks regarding the scope of our contributions and writing. Below, we address the requested changes and suggestions.
>
> > 1. Table 3, right: Why the sudden increase in planning time at level 3?
>
> It happens that the first level is most difficult in our Sokoban version. Since this is confusing, we reordered Sokoban’s table. Now, it can be seen that the planning time should increase as you go up in level since in the later levels you need to push more boxes around. In push boulders 1, that is why there is increased planning time in level 3 since it is more difficult than the first two levels.
>
> > 2. ... Why not just include the time and then mention why their values are the same?
>
> Thank you for this suggestion. We included the time and mentioned why they are the same.
>
> > 3. .. It would be useful to include more tasks where TheoryCoder has substantial room for improvement.
>
> Thank you for this important feedback. We evaluated our system on more challenging levels outside of the demo levels in KekeCompetition and identified areas of substantial improvement for our system. We included the result and analysis in section 4.7, accompanied by Figure 14. We found two tasks where TheoryCoder has substantial room for improvement and fails even after 25 API calls. In these tasks, the agent needs to challenge its assumptions very rigorously by sacrificing two of the controlled characters into the goop to intentionally die to clear a path. This highlights a broader area for improvement which is preventing overgeneralization from early transitions. We suggest that this can be mitigated in future work by extending TheoryCoder to build a set of multiple world models and cycling between them when one repeatedly fails. The agent can also be prompted or encouraged to do this type of flexible thinking.
>
> > 4. How does theory-based RL relate to the options literature [1]?
>
> Thank you for this insight, this is a good question. We mean to convey that our findings provide additional support for this claim: “stratified learning strategies applied to hierarchical abstractions can improve planning and learning efficiency in non-trivial problem domains”. We acknowledge that options and hierarchical RL have argued for similar points. Our work shares the philosophy of the options framework in that it is a stratified learning strategy, however we develop a different approach to learn and represent abstractions based on program synthesis. We have also edited our conclusion in our revision to cite the options literature: “Overall, our findings support the hypothesis that stratified learning strategies applied to hierarchical abstractions can improve planning and learning efficiency \citep{sutton1999between} in non-trivial problem domains like Baba is You.”
>
> > **(Requested changes)** While the paper isn't difficult to parse, readability would be improved if the authors added additional headers. As an example, the related work ...
>
> Thank you for this suggestion, we feel that it really helped improve the readability of our related work section. We have broken it up with multiple bold headers.
>
> > **(Requested changes)** “Second, the planning algorithms used in previous TBRL agents are not scalable to large state spaces.” Brief describe why.
>
> We have included the following explanation: They are not scalable since they rely on value-based iteration, which suffers from the curse of dimensionality when maintaining value estimates over large state-action spaces (Sutton et al., 1998).
>
> > **(Requested changes)** Section 1: the meaning of "error distribution" is unclear.
>
> We have removed the word “distribution”, so now it reads:  “However, the errors they make are also very unlike humans’. For example, they have a poor sense of the bounds of their knowledge and competencies, which leads to overconfidence when encountering situations far from their training sets.”
>
> > **(Requested changes)** Section 1: clearly describe what is meant by "in-context" here
>
> We added the description: “..’in-context’ learning Russin et al. (2024) which is defined as being able to solve new tasks without fine-tuning, by being provided with a few examples”
>
> > **(Requested changes)** Section 3.2: “Thus, the optimal plan is the shortest sequence of actions from s1 to sN = s∗.” nitpick: an optimal plan is the shortest action sequence that reaches the goal. If you include discounting, then it is the optimal plan.
>
> Thank you for finding this detail! We made this change in our revision.
>
> > **(Requested changes)** Algorithm 1, line 9 ...
>
> We removed the line in the algorithm that sampled from the world model, so that it now correctly conveys execution of actions in the environment.
>
> > **(Requested changes)** ... mention that 4o is less powerful than o1 in Section 4.1.
>
> We added this change to section 4.1 as suggested.
>
> To conclude, we thank the reviewer again for their detailed feedback and suggestions, which have improved our writing.

---

### Review · Reviewer_Yy8F · 2025-05-29

**Summary Of Contributions:**

In this paper the authors present an approach that combines LLMs and bilevel planning. They use a domain-specific PDDL setup to create a high-level plan and then use an LLM prompt to construct a low-level world model, with the potential to use an update to ask the LLM to update the low-level world model, which they can then plan in. They demonstrate that the approach can work across three small domains and outperforms an approach to extract a plan by just prompting the LLM.

**Audience:**

Yes

**Broader Impact Concerns:**

I don't have any concerns around broader impacts.

**Claims And Evidence:**

No

**Requested Changes:**

I'd suggest changes around each of my concerns from above:
1. Testing the approach in an environment that that the LLM used could not have been trained on would be ideal. Further, explaining how TheoryCoder could have 0 API usages would be beneficial.
2. Adding additional relevant baselines, but especially EMPA given the claims around choosing Sokoban and Push Boulders 1 due to EMPA's poor performance.
3. Cutting back on the length of the writing throughout the first half of the paper. As an example, in section 3.3.1 I estimate the authors could completely cut the first three paragraphs and a majority of the fifth and seventh paragraphs.
4. Provide a systematic experiment around transferability or cut the related claims.

**Strengths And Weaknesses:**

The primary strength of the paper is in the concept for integrating LLMs, bilevel planning, and world models. This is, to the best of my knowledge, a novel concept and could potentially lead to improvements in automated game playing.

I am concerned with a number of weaknesses of the current paper.
1. Validity - The authors make use of GPT-4o, which was updated based on a scrape in January 2022, which is well after the date that the earliest versions of the codebases for all of the test environments were publicly available online. This means that GPT-4o may have been trained on these codebases, which would not provide information for how GPT-4o would perform in unseen environments. This could also explain why the authors' approach requires in some cases 0 API uses.
2. Lack of Baselines - The authors only compare against a single baseline of prompting an LLM and an ablation of only using low-level planning. There is no comparison to EMPA, other automated game playing approaches, pure planning approaches, other world model approaches, or other world model learning + model-based RL approaches (e.g., "Guzdial, Matthew, Boyang Li, and Mark O. Riedl. "Game Engine Learning from Video." IJCAI. 2017.)
3. Writing - The paper is currently very long, and much of the first half of the paper has surprisingly introductory text, which is not appropriate for a TLMR paper.
4. Transferability Claims Support - The authors' only proof when it comes to the transferability is a single visual example with no deeper analysis or broader evidence.

---

> ### Author Response · Authors · 2025-06-05
> **Response to Reviewer Yy8F**
>
> We sincerely thank the reviewer for the valuable time reviewing our work. We appreciate their encouraging comments on the novelty of our approach. We clarify and address the reviewer's concerns as follows.
>
> > **(Requested changes)** Testing the approach in an environment that that the LLM used could not have been trained on would be ideal. Further, explaining how TheoryCoder could have 0 API usages would be beneficial.
>
> Thank you for bringing up this insight. To address this, we design a new game environment and report the results in section 4.9 and Figure 15. We find that our approach still has the same behavior of being able to learn from few interactions and solve similar problems once it learns the underlying concept. We would also like to mention that although the KekeCompetition codebase was public at that time, the code was available in Javascript and we wrote our own game engine in python for it. Finally, knowledge of the game engine may not be that useful since learning a transition model code is quite different from the raw game engine code.
>
> To clarify, 0 API uses means that our agent is able to do planning offline to find the solution, without needing to call the LLM. For example, if our agent uses 1 API call to learn a line of code that represents colored doors that can be unlocked by keys of the same color, then it can solve any level where it needs to unlock doors of different colors without needing to call the LLM to learn any new concept. Our tables illustrate this trend of learning and reusing the knowledge offline.
>
> TheoryCoder having 0 API usage is beneficial since cost can be rapidly incurred through LLMs especially LLM based agents such as reasoning models. The experiments with o1 in just these small domains incurred around $150-200 dollars in cost. Of course there is a tradeoff that one must consider since planning offline can also be slow during runtime and we acknowledge that there is usefulness to hybrid systems where LLMs are called to speed up planning and external planning is used to augment LLMs shortcomings.
>
> We have also included this explanation for 0 API calls in section 4.3 page 16 and explained why minimizing API usage is important in section 4.5’s beginning . We had already discussed the trade off in the second to last paragraph of section 4.5.
>
> > **(Requested changes)** Adding additional relevant baselines, but especially EMPA given the claims around choosing Sokoban and Push Boulders 1 due to EMPA's poor performance.
>
> Thank you for these valuable comparison suggestions. We have included the following relevant baselines:
>
> - A comparison to EMPA in section 4.3 Table 3 and Figure 6, where we show that TheoryCoder’s learning efficiency shows significant improvement in Sokoban and Push Boulders 1, where EMPA only achieves 50% success rate in Push Boulders 1. In Sokoban, it can be seen that TheoryCoder is able to achieve 100% success in almost half the number of steps in the environment.
>
> - We compare the computational efficiency of the pure planning approach of using Fast Downward directly in section 4.3 Table 4 and find our approach to be much more efficient.
>
> - We compare to another world model code learning approach (WorldCoder) in section 4.3 Table 7 and again find our system is able to achieve greater success rate with significantly less total tokens cost.
>
> We left out a comparison to a model-based RL approach directly since EMPA and WorldCoder already showed substantial efficiency over these approaches. However, we are open to further discussion if concerns still exist.
>
> The game engine learning from video paper is interesting, however the setting is different from ours. All of our relevant baselines operate directly on symbolic descriptions, however that approach learns from raw pixels. We do believe that this could be interesting future work to incorporate similar types of learning directly from pixel input, and we have cited the paper in our related work.
>
> > **(Requested changes)** Cutting back on the length of the writing throughout the first half of the paper. As an example, in section 3.3.1 I estimate the authors could completely cut the first three paragraphs and a majority of the fifth and seventh paragraphs.
>
> Thank you for pointing this out. We cut the majority of the first three paragraphs and fifth and seventh paragraphs.
>
> > **(Requested changes)** Provide a systematic experiment around transferability or cut the related claims.
>
> Thank you for this suggestion. We cut the claims about transferability and removed that section. We agree a systematic deeper analysis would be necessary to draw any conclusions. We do want to mention that it could be an interesting future direction due to the potential of code to be transferable.
>
> To conclude, we thank the reviewer again for their valuable comments and suggestions, which have improved the quality of our work. We hope our response addresses the concerns, and we would be glad to discuss further.

---

> > ### Comment · Reviewer_Yy8F · 2025-06-10
> > **RE: Response to Reviewer Yy8F**
> >
> > Thanks to the authors for their comprehensive response! I've reviewed everything and I am pleased with the response to the majority of my requested changes (1, 2, and 4). However, I think I may have poorly expressed my concern around the third requested change (3). Section 3.3.1 was meant as an example but I think there are places to tighten up the writing in the first three sections of the paper. If the authors wish I can give more specific feedback, but generally I was concerned by how broad the writing is, for example explaining quite a bit about human psychology in a depth not necessary to understand the authors' work.

---

> > > ### Author Response · Authors · 2025-06-11
> > > **Response to Comment by Reviewer Yy8F**
> > >
> > > Thank you very much — we really appreciate your response and feel that your feedback has helped us improve the paper significantly.
> > >
> > > For your third requested change regarding the writing, we have further trimmed portions from the first three sections — specifically: the second and sixth paragraphs of the introduction, the first paragraph of Section 3.1, and the third paragraph of Section 3.4. The text we intend to cut is marked with red strikethrough in the revision.
> > >
> > > We are considering retaining some of the discussion around human psychology, as we believe it helps motivate our framing of theory-based RL. So for now, we’ve been more conservative in removing those parts. That said, we’re very open to further suggestions and would welcome any specific feedback you may have to tighten the writing further.
> > >
> > > Thank you again for your thoughtful feedback!

---

### Author Response · Authors · 2025-06-23
**End of discussion period is imminent & Revision uploaded**

This is a friendly reminder that there are only 3 days left for the discussion period. In case the reviewers have any remaining concerns, please let us know, we would love to clarify more. Otherwise, we'll assume that our responses have successfully addressed all the concerns raised. We have updated our revised manuscript accordingly. The list of changes can be found above.

We sincerely thank all the reviewers once again for reviewing our work and giving valuable feedback.

---

### Decision · Action_Editor_wPBo · 2025-06-27

**Recommendation:** Accept as is

**Additional Comments:**

The paper introduces TheoryCoder, a theory-based reinforcement learning algorithm that uses LLMs for synthesizing Python-based world models and integrates these with bilevel planning via PDDL. The work shows strong empirical results and also outperfroms newer models like GPT4-o which is an interesting result. The paper excels in demonstrating strong sample efficiency and a clear separation between abstract reasoning and grounded execution making the framework modular in nature which can thus be easily verifiable and adaptable. Although, I do think that the choice of datasets could have been expanded to show the effectivieness of the algorithm on a more complex, high dimensional environment(s) but this can be deferred to future work since some modifications to the current setup will be required.

Overall, along with the reviewers I believe that this is a strong contribution that warrants acceptance. I would like to request the authors to also take the reviewers comments into account (since most points are very minor I am recommending an as-is accept) when preparing the final draft. Congratulations on the great work.

**Audience:**

Yes

**Audience Explanation:**

Yes. A broad set of audience will be interested in this work. The work nicely combines the modern aspects of LLM based program synthesis with the classical aspects of bilevel planning to report an advancement in the area of theory based reinforcement learning. I think a lot can be taken from this paper for audience from several areas of machine learning.

**Claims And Evidence:**

Yes

**Claims Explanation:**

Yes, the claims in the submission seem accurate and are supported by clear empirical evidence. All the reviewers support this.

---

> ### Author Response · Authors · 2025-07-11
> **Thank you and camera-ready uploaded**
>
> Dear Action Editor and Reviewers,
>
> Thank you for your favorable final decision. We have uploaded the camera-ready version.
>
> We greatly appreciate the time and effort you invested in reviewing our paper and providing valuable feedback. Thank you very much once again.
>
> Best regards,
>
> Authors